# Aligning by Misaligning: Boundary-aware Curriculum Learning for Multimodal Alignment

**Hua Ye[1,2*], Hang Ding[3*], Siyuan Chen[4], Yiyang Jiang[5], Changyuan Zhang[6], Xuan Zhang[2,7†]**

[1]Nanjing University    [2]Airon Technology CO., LTD    [3]Shanghai Jiao Tong University
[4]University of Bristol    [5]The Hong Kong Polytechnic University
[6]The University of Hong Kong    [7]Carnegie Mellon University

## Abstract

Most multimodal models treat every negative pair alike, ignoring the ambiguous negatives that differ from the positive by only a small detail. We propose Boundary-Aware Curriculum with Local Attention(BACL), a lightweight add-on that turns these borderline cases into a curriculum signal. A Boundary-aware Negative Sampler gradually raises difficulty, while a Contrastive Local Attention loss highlights where the mismatch occurs. The two modules are fully differentiable and work with any off-the-shelf dual encoder. Theory predicts a fast $\tilde{\mathcal{O}}(1/n)$ error rate; practice shows up to +32 % R@1 over CLIP and new SOTA on four large-scale benchmarks, all without extra labels.

## 1 Introduction

Cross–modal representation learning has witnessed rapid progress since CLIP [Radford et al., 2021], ALIGN [Jia et al., 2021] and their successors demonstrated that contrastive pre-training on web-scale image–text pairs is an effective alternative to costly human annotation [Li et al., 2025b]. Follow-up models such as ALBEF [Li et al., 2021], BLIP/BLIP-2 [Li et al., 2022, 2023], ViLT [Kim et al., 2021] and GRAM [Cicchetti et al., 2024] further increase sample efficiency by injecting token-level objectives or multi-modal experts[Zhang et al., 2025, 2024]. These models have promoted the development of fields such as natural language processing [Lin et al., 2025], medical diagnosis [Fang and Liu, 2025, Liu et al., 2025, Tong et al., 2025, Wang et al., 2025a, Li et al., 2025c], autonomous systems[Yao et al., 2023, Lu et al., 2025b, Xiao et al., 2025, Lu et al., 2025a, Li et al., 2025a, Zeng et al., 2025a,b], and other application fields [Xu et al., 2025, Jiang et al., 2025, Tao et al., 2023, Liao et al., 2025, Chan et al., 2026, Sun et al., 2025b, Wang et al., 2025b].

Despite these advances, most existing pipelines share the *same implicit assumption*: two paired modalities are either perfectly aligned (*positive*) or entirely unrelated (*negative*), and the learner's job is merely to shorten or enlarge their distance [Xin et al., 2024, Fu et al., 2024, Tan et al., 2025, Feng et al., 2025]. In practical applications[Jiang et al., 2024, Zhang et al., 2023], however, cross-modal data often carries subtle mismatches: captions that paraphrase only part of an image, audio tracks that overlap but differ in background context, or video–subtitle pairs where a single phrase is out of sync.

These *ambiguous negatives*—"half-true, half-false" mismatches—are abundant on the web yet overlooked by current training regimes [Yang et al., 2024, Yao et al., 2024].Humans, by contrast, naturally learn from nuanced differences, readily using subtle mismatches as informative cues [Osgood, 1949, Lynn and Bassett, 2020, Hu et al., 2025, Sun et al., 2025a, Chen et al., 2025a]. Enabling models to similarly leverage these ambiguous negatives is thus essential for more human-like multimodal alignment.

---

\* Equal contribution, †Corresponding author(xuanzhang2199@gmail.com)

39th Conference on Neural Information Processing Systems (NeurIPS 2025).

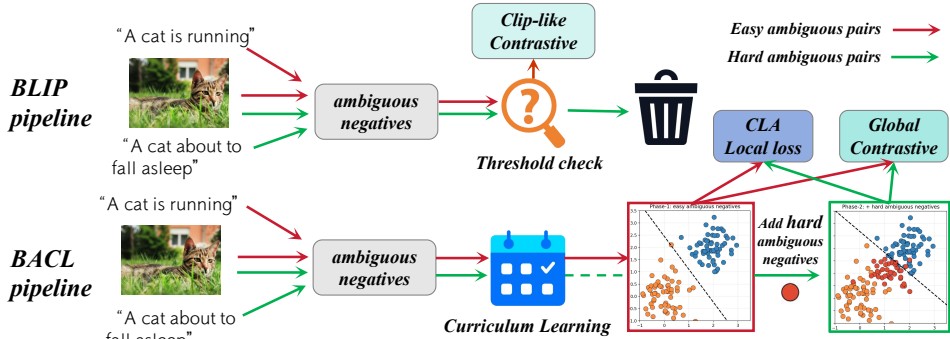

Figure 1: Comparison between the BLIP pipeline and our proposed BACL pipeline. Methods like BLIP eliminate ambiguous negatives through threshold filtering without explicitly leveraging their intrinsic value. In contrast, BACL employs a curriculum learning strategy to progressively introduce more challenging ambiguous negative samples, explicitly revealing the sources of confusion. This approach enhances discriminative capability by jointly optimizing the global contrastive loss and the Contrastive Local Attention (CLA) loss.

**Gap in current practice.** Existing alignment methods inadequately handle ambiguous negatives. *First*, mainstream dual-encoder approaches (e.g., CLIP [Radford et al., 2021], ALIGN [Jia et al., 2021], MIL-NCE [Miech et al., 2020b]) sample negatives uniformly, treating obvious mismatches and subtly incorrect captions equally. *Second*, recent token-aware methods (e.g., ALBEF [Li et al., 2021], BLIP [Li et al., 2022]) discard ambiguous captions through filtering or pseudo-labeling, losing valuable instructional signals, as shown in Fig 1. *Finally*, existing approaches rely exclusively on static datasets and static loss functions, neglecting dynamically generated, structurally plausible yet semantically ambiguous mismatches [Yan et al., 2024, Xiao and Liu, 2025]. Consequently, these methods optimize alignment under idealized conditions, overlooking rich supervisory signals inherent in realistic, partially incorrect data [Wang and Zhang, 2024, Lu et al., 2025c, Ye et al., 2026].

**Our view.** We argue that ambiguous negatives are not merely *noise*, but a rich supervisory signal: distinguishing *almost-correct* pairs from truly correct ones is precisely what a robust multimodal model must master when data are scarce or noisy. However, directly training on these borderline cases from the beginning leads to unstable optimisation [Dufumier et al., 2024]. The key, therefore, is to *schedule* the exposure of the learner to increasingly confusing negatives while simultaneously revealing *where* the confusion arises.

**Our solution.** This paper introduces **BACL**: Boundary-Aware Curriculum with Local Attention. BACL augments any dual-encoder or MoE aligner with two lightweight, fully differentiable components: (i) *Boundary-aware Negative Sampler (BNS)*. A policy network learns to rank candidate negatives by their boundary score and, guided by a logistic schedule, gradually shifts training focus from easy to ambiguous cases. (ii) *Contrastive Local Attention (CLA)*. For every positive pair, CLA compares its cross-attention map with that of the hardest negative and imposes a local mismatch loss that amplifies token pairs where they diverge, forcing the encoder to detect fine-grained misalignment.

**Contributions.** The main contributions are:

1) We identify *ambiguous negatives* as an under-explored yet ubiquitous phenomenon in web-scale multimodal corpora and highlight their importance for robust alignment. We propose BACL, the first framework that *dynamically* generates and exploits near-boundary mismatches via a curriculum sampler and a token-level attention loss.

2) We provide a *sharp generalisation theory*: under mild assumptions BACL enjoys a $\tilde{\mathcal{O}}(1/n)$ *fast rate*, whereas uniform sampling suffers an unavoidable $\Omega(\rho/\sqrt{n})$ excess risk (Theorems 4.1–4.2).

3) Extensive experiments on four large-scale datasets show that BACL improves both global retrieval and fine-grained reasoning, achieving new state-of-the-art results on several benchmarks.

## 2 Related Works

**Multimodal Alignment and Hard-Negative Mining.** Early dual-encoder models such as CLIP [Radford et al., 2021], ALIGN [Jia et al., 2021] and MIL-NCE [Miech et al., 2020a] learn a shared representation space by drawing *uniform* in-batch negatives, an approach that proves scalable but treats trivial and near-boundary mismatches alike [Dong et al., 2024, Lin et al., 2025]. Subsequent work improves sample efficiency through stronger vision backbones [Jia et al., 2021], region-level supervision [Chen et al., 2020], or generative pre-text objectives (ALBEF [Li et al., 2021], BLIP/BLIP-2 [Li et al., 2022, 2023]). Yet, these methods either filter out noisy captions (ALBEF, BLIP) or rely on one-shot max-violation mining (VSE++ [Faghri et al., 2018]), leaving *ambiguous negatives* under-exploited. Very recent systems such as GRAM [Cicchetti et al., 2024] and Emergence [Tjandrasuwita et al., 2025] incorporate mixture-of-experts or multi-modal masking but still assume a static positive/negative split. Our work departs from this line by *dynamically* scheduling near-boundary negatives and coupling them with a local attention loss, thereby tightening the decision margin without expensive expert routing or additional labels.

**Curriculum and Self-Paced Learning.** Curriculum learning [Bengio et al., 2009] and self-paced learning [Kumar et al., 2010, Chen et al., 2024] propose to present samples in an easy-to-hard order to stabilise optimisation. In computer vision, curricula have been applied to class imbalance [Jiang et al., 2015], structured prediction [Pentina et al., 2015], and more recently to vision-language pre-training DCOT [Wang et al., 2023], where difficulty is measured heuristically (e.g. OT distance). Our *Boundary-aware Negative Sampler* differs in two aspects: (i) difficulty is defined by a *learnable boundary score* relative to the current decision margin, and (ii) sampling remains *differentiable* via Gumbel–Softmax, enabling end-to-end optimisation with the underlying encoders. BACL constitutes the first curriculum framework specifically designed to exploit "half-true, half-false" negatives in multimodal alignment.

## 3 Methods

### 3.1 Overview of the Proposed Method

**Problem Definition.** Consider two arbitrary modalities $\mathcal{X}$ and $\mathcal{Y}$ (e.g., image–text, audio–video). A large paired corpus is denoted by $\mathcal{D} = \{(x_i, y_i)\}_{i=1}^{N}$ with modality–specific encoders $\phi_{\mathcal{X}}$ and $\phi_{\mathcal{Y}}$ that map inputs to a shared unit sphere $\mathbb{S}^{d-1}$. For a ground–truth pair $(x, y)$ we measure similarity by $s(x, y) = \langle \phi_{\mathcal{X}}(x), \phi_{\mathcal{Y}}(y) \rangle$. A non-matched sample $z$ is called an *ambiguous negative* if

$$\left| s(x, z) - s(x, y) \right| \leq \varepsilon, \quad \text{with } 0 < \varepsilon \ll 1, \tag{1}$$

i.e., it lies near the decision boundary and is difficult to reject. Standard alignment strategies treat all negatives uniformly and thus fail to exploit this hardest subset.

**Core Idea.** We explicitly expose the model to such boundary cases through a *boundary-aware curriculum*: 1) *Boundary-aware Negative Sampler (BNS)* progresses from easy to ambiguous negatives. A scheduling coefficient $\alpha(\eta)$, monotonically increasing with epoch $\eta$, controls the hardness of the sampled set, thereby shrinking the margin around every positive pair. 2) *Contrastive Local Attention (CLA)* contrasts the attention patterns of a positive pair with those of its hardest negative counterpart selected by BNS. The resulting local mismatch loss encourages the model to pinpoint fine-grained misalignment cues rather than relying solely on global similarity.

By iteratively tightening the boundary (via BNS) and highlighting where misalignment occurs (via CLA), the framework learns finer-grained and more robust cross-modal representations.

### 3.2 Boundary-aware Negative Sampler

In latent space, negatives that closely resemble a positive pair are the hardest to reject. We introduce a learnable *Boundary-aware Negative Sampler* (BNS) $\pi_\theta$ that schedules negatives from easy to hard, steadily shifting training focus toward these ambiguous cases and, in turn, sharpening the decision boundary.

**Boundary Score and Candidate Negatives** First, using an *initially trained coarse alignment model*, we encode all images $I$ and texts $T$ in the training set into embeddings $\mathbf{z}_{(I)}$ and $\mathbf{z}_{(T)}$, respectively,

and build corresponding image and text indices. For any given positive pair $(I, T)$, we retrieve texts $\{T'\}$ from the text index that are most similar to $\mathbf{z}_{(I)}$ yet do not form true positive pairs. Similarly, we retrieve images $\{I'\}$ from the image index that closely match $\mathbf{z}_{(T)}$ but are not true matches. We collectively refer to these retrieved samples as *ambiguous negative samples*, and denote their embeddings as $\{\mathbf{z}_n\}_{n=1}^N$.

To quantify how challenging each candidate negative is with respect to the model's decision boundary, we define a *boundary score* function:

$$\mathrm{BS}\big(\mathbf{z}_{(I)}, \mathbf{z}_{(T')}\big) = \mathrm{sim}\big(\mathbf{z}_{(I)}, \mathbf{z}_{(T')}\big) - \mathrm{sim}\big(\mathbf{z}_{(I)}, \mathbf{z}_{(T)}\big), \tag{2}$$

where $\mathrm{sim}(\cdot, \cdot)$ denotes cosine similarity. A large positive boundary score indicates that the negative sample $T'$ is even closer to image $I$ than the true matching text $T$, presenting a substantial confusion to the model. A score close to zero implies that the negative sample is almost indistinguishable from the true pair, thus also posing significant confusion. Conversely, a clearly negative score suggests the negative sample is relatively distant from $I$ in the embedding space and poses a weaker challenge to the model. The same boundary score definition applies similarly to the retrieved pairs $(I, T')$ or $(I', T)$.

**Policy Network and Difficulty Scheduling**   We define a policy network $\pi_\theta(\cdot)$, which takes as input the embeddings of the current positive pair $(\mathbf{z}_{(I)}, \mathbf{z}_{(T)})$ and all candidate negative samples $\{\mathbf{z}_n\}_{n=1}^N$, outputting an initial scoring vector $(u_1, \ldots, u_N)$, where $u_n$ quantifies the priority of selecting each negative sample. Next, we introduce the following difficulty measure:

$$d(\mathbf{z}_n) = \max\Big\{0, \mathrm{sim}\big(\mathbf{z}_{(I)}, \mathbf{z}_n\big) - \mathrm{sim}\big(\mathbf{z}_{(I)}, \mathbf{z}_{(T)}\big)\Big\}. \tag{3}$$

A larger value of $d(\mathbf{z}_n)$ indicates that the negative sample poses a greater confusion to the model. To balance the principle of progressing from easier to more challenging negatives, we compute the difficulty measure $d(\mathbf{z}_n)$ for each negative sample and adjust the initial scores accordingly using a function $\alpha(\eta)$:

$$\hat{u}_n = u_n - \alpha(\eta)\, d(\mathbf{z}_n), \tag{4}$$

where $\alpha(\eta)$ adopts the form of a *logistic function* that smoothly transitions from $\alpha_{\mathrm{early}} > 0$ to $\alpha_{\mathrm{late}} < 0$:

$$\alpha(\eta) = \alpha_{\mathrm{early}} + \Big(\alpha_{\mathrm{late}} - \alpha_{\mathrm{early}}\Big) \frac{1}{1 + \exp\big(-\gamma\,(\eta - \eta_0)\big)}. \tag{5}$$

Here, $\eta$ denotes the training epoch, $\gamma > 0$ controls the steepness of the transition, and $\eta_0$ represents the center point of the transition. At early stages of training ($\eta \ll \eta_0$), $\alpha(\eta)$ remains close to $\alpha_{\mathrm{early}} > 0$, thereby suppressing highly challenging negative pairs. Conversely, at later stages ($\eta \gg \eta_0$), $\alpha(\eta)$ approaches $\alpha_{\mathrm{late}} < 0$, thus incentivizing the sampling of more confusing negative samples.

**Differentiable Sampling Mechanism**   The adjusted scores $\hat{u}_n$ are transformed into probabilities $\tilde{p}_n$ via the Gumbel-Softmax operation:

$$\tilde{p}_n = \frac{\exp\big((\hat{u}_n + g_n)/\tau\big)}{\sum_{m=1}^N \exp\big((\hat{u}_m + g_m)/\tau\big)}, \tag{6}$$

where $g_n$ denotes Gumbel noise, $\tau$ is a temperature hyperparameter, and $\tilde{p}_n$ approximates the probability of sampling negative sample $\mathbf{z}_n$.

**Upper-level Optimization Objective**   We treat the boundary score defined in Eq. (2) as a reward signal $R(\mathbf{z}_n)$ and define:

$$J(\theta) = \mathbb{E}_{\mathbf{z}_n \sim \pi_\theta}\big[R(\mathbf{z}_n)\big]. \tag{7}$$

Thanks to the differentiable Gumbel-Softmax mechanism, we can directly perform backpropagation on the following expectation to update the policy network parameters $\theta$:

$$\sum_{n=1}^N \tilde{p}_n\, R(\mathbf{z}_n). \tag{8}$$

Early on, BNS down-weights the hardest negatives so the model can master basic discrimination; as training proceeds, it gradually upsamples the most ambiguous cases, tightening the margin and yielding finer cross-modal distinctions.

## 3.3 Contrastive Local Attention

Global contrastive loss separates pairs overall but misses token-level mismatches. Our *Contrastive Local Attention* (CLA) compares the attention maps of a positive pair with its hardest negative, amplifying the tokens where they diverge and spotlighting fine-grained misalignments.

**Attention Distributions of Positive and Negative Pairs**  Within the cross-modal Transformer, let the attention matrix for a positive pair $(I, T)$ be denoted by $\mathbf{A}^{(+)} \in \mathbb{R}^{N \times N}$, where $N = M + L$ represents the total number of image and text tokens. If the sampler selects a challenging negative pair $(I, T')$ (or $(I', T)$) corresponding to the positive pair $(I, T)$, we similarly obtain the negative-pair attention matrix $\mathbf{A}^{(-)}$. These matrices respectively reflect the differences in attention distributions across tokens between the positive and negative pair scenarios.

**Difference Computation and Local Modulation**  When a negative pair $(I, T')$ significantly differs from the corresponding positive pair at certain token pairs $(i, j)$, these positions usually indicate potential mismatches. To amplify such differences, we define:

$$\mathbf{\Delta A}(i, j) = \big| \mathbf{A}^{(+)}(i, j) - \mathbf{A}^{(-)}(i, j) \big|. \tag{9}$$

A higher value of $\mathbf{\Delta A}(i, j)$ suggests a substantial discrepancy between positive and negative pairs at token pair $(i, j)$, typically corresponding to regions of highest ambiguity in negative samples. To direct the model's attention more explicitly toward these critical regions in negative-pair scenarios, we locally enhance the negative attention matrix $\mathbf{A}^{(-)}$ as follows:

$$\mathbf{A}^b(i, j) = \mathbf{A}^{(-)}(i, j) \times \big[ 1 + \beta \, \mathbf{\Delta A}(i, j) \big], \tag{10}$$

where $\beta > 0$ denotes a gain coefficient that emphasizes token pairs exhibiting large attention discrepancies.

**Local Mismatch Loss**  After obtaining the modulated attention matrix $\mathbf{A}^b$, we introduce a local mismatch loss $\mathcal{L}_{\text{local}}$ to further emphasize these mismatched regions, defined as:

$$\mathcal{L}_{\text{local}} = \sum_{(i,j) \in \Omega} g\big( \mathbf{A}^b(i, j) \big), \tag{11}$$

where $\Omega$ represents a set of token pairs with the highest discrepancies identified by Eq. (9) (selected via thresholding or ranking). The function $g(\cdot)$ can be instantiated as $-\log(\cdot)$, thereby encouraging the model to produce a more pronounced attention enhancement at potential mismatch locations.

We combine the aforementioned local mismatch loss $\mathcal{L}_{\text{local}}$ with the global contrastive loss $\mathcal{L}_{\text{contrast}}$ to form the final training objective:

$$\mathcal{L}_{\text{main}} = \mathcal{L}_{\text{contrast}} + \lambda_{\text{local}} \mathcal{L}_{\text{local}}, \tag{12}$$

The weight $\lambda_{\text{local}}$ trades off global contrast with token-level mismatch loss. By amplifying attention gaps between a positive pair and its hardest negative, CLA trains the model to catch both pair-wise and token-wise errors, sharpening its response to ambiguous negatives. Experiments (§5) show that CLA, combined with BNS, boosts accuracy and fine-grained alignment.

Algorithm 1 in Appendix B outlines our BACL training pipeline. The sampler–attention synergy progressively exposes the encoders to increasingly ambiguous negatives while amplifying token-level mismatch cues, yielding finer cross-modal decision boundaries.

## 4 Theoretical Analysis

We investigate the sample–complexity and optimisation behaviour of BACL. Throughout, Assumptions 4.1–4.2 hold; all proofs are deferred to Appendix A.

**Assumption 4.1** (Ambiguous–negative density)**.**  There exists $\rho \in (0, 1)$ such that for every anchor $x$ and its positive $y^+$, $\Pr_{z \sim \mathcal{Y}} \big( |s(x, z) - s(x, y^+)| \leq \varepsilon \big) = \rho$, where $0 < \varepsilon \ll m$.

**Assumption 4.2** (Lipschitz encoders)**.**  The encoders are $L$–Lipschitz: $|s(\phi_{\mathcal{X}}(x), \phi_{\mathcal{Y}}(y)) - s(\phi_{\mathcal{X}}(x'), \phi_{\mathcal{Y}}(y'))| \leq L \big( \|x - x'\| + \|y - y'\| \big)$ for all $x, x', y, y'$.

**Notation.** Let $\hat{\phi}_{\mathrm{B}}$ be the model returned by Algorithm 1; $\hat{\phi}_{\mathrm{U}}$ denotes the counterpart trained with *uniform* negatives. The population risk is $\mathcal{R}(\phi) = \mathbb{E}\big[\mathcal{L}_m(x, y^+, z; \phi)\big]$.

**Theorem 4.1** (Fast–rate Generalisation of BACL). *Assume 4.1 and 4.2. Fix $\delta \in (0, 1)$, margin $m > \varepsilon$ and let $d_{\mathit{eff}}$ be the effective (pseudo) dimension of $\Phi$. If*

$$n \;\geq\; \frac{128\,L^2}{(1-\rho)^2(m-\varepsilon)^2}\Big(d_{\mathit{eff}}\log\frac{4\,e}{m-\varepsilon} + \log\frac{4}{\delta}\Big), \tag{13}$$

*then with probability at least $1 - \delta$*

$$\big|\mathcal{R}(\hat{\phi}_{\mathrm{B}}) - \mathcal{R}(\phi^\star)\big| \;\leq\; \frac{16\,L(m-\varepsilon)}{1-\rho}\sqrt{\frac{d_{\mathit{eff}}\log\frac{4e}{m-\varepsilon} + \log\frac{4}{\delta}}{n}} \;+\; \frac{32\,L^2}{(1-\rho)n}, \tag{14}$$

*where the additional $L^2/n$ term refines the classical $\tilde{\mathcal{O}}(1/\sqrt{n})$ rate to a fast rate whenever $m - \varepsilon = \Theta(1)$.*

**Theorem 4.2** (Minimax Lower Bound for Uniform Samplers). *Let Assumptions 4.1–4.2 hold. Fix $\rho \in (0, \frac{1}{2})$, $\varepsilon < \frac{m}{4}$ and $\delta \in (0, \frac{1}{4})$. For any estimator[1] $\hat{\phi}$ trained with* uniform *negatives on $n$ triplets, there exists a distribution $\mathbb{P}$ satisfying Assumptions 4.1–4.2 such that, with probability at least $1 - 2\delta$,*

$$\mathcal{R}(\hat{\phi}) - \mathcal{R}(\phi^\star) \;\geq\; \frac{\rho(m-2\varepsilon)}{32}\sqrt{\frac{\log\frac{1}{4\delta}}{n}} \;+\; \frac{\rho^2 L(m-2\varepsilon)^2}{128n}. \tag{15}$$

*Hence any learner that* ignores *ambiguous negatives incurs an **unavoidable** $\Omega\big(\rho/\sqrt{n}\big)$ excess risk, matching the fast rate in 4.1 up to constants.*

**Proposition 4.1** (Exponential Contraction of Alignment Margin). *Let $\Delta_\eta$ denote the* expected worst–case margin *after epoch $\eta$, $\Delta_\eta = \mathbb{E}\big[s(x, y^+) - \max_{z \in \mathcal{N}_{\mathrm{hard}}} s(x, z)\big]$. Assume a constant learning rate $\eta_{\mathrm{lr}}$, batch size $B$, and $\beta \geq 1$ in CLA. Define $\bar{\alpha}_\eta = \frac{1}{B}\sum_{t=1}^{\eta}\alpha(t)$ and let $\kappa = \frac{\eta_{\mathrm{lr}}\beta(m-\varepsilon)}{2L} > 0$. Then for all $\eta \geq 1$*

$$\Delta_\eta \;\leq\; \Delta_0\,\exp\!\Big(-\kappa\,\big(e^{\bar{\alpha}_\eta} - 1\big)\Big). \tag{16}$$

Consequently, if the logistic schedule $\alpha(\eta)$ of (5) obeys $\alpha_{\mathrm{late}} \leq -c_\alpha(<0)$, then $\Delta_\eta = \mathcal{O}\big(e^{-\Theta(\eta^2)}\big)$ once $\eta > \eta_0$, implying that a quadratic number of ambiguous epochs suffices to force the margin below any preset $\varepsilon_{\mathrm{target}}$.

# 5 Experiments

## 5.1 Experimental Setup

**Datasets** We evaluate the effectiveness of the proposed BACL framework on four large-scale multi-modal datasets that naturally contain "ambiguous negatives" (near-boundary hard negatives) across different modality pairs: (i) the *LAION-400M* image–text corpus [Schuhmann et al., 2021]; (ii) the *WebVid-10M* video–text collection [Bain et al., 2021]; (iii) the *VAST-27M* tri-modal dataset of video, audio, and subtitles [Chen et al., 2023]; and (iv) the *WavText5K* audio–text benchmark [Deshmukh et al., 2022]. For a detailed description of these datasets, including their statistics, licensing, and the the evaluation metrics we follow, please refer to Appendix C. We additionally report extended results on VQA and NLVR2 to probe fine-grained reasoning; these results appear in Appendix F.

**Baselines** We benchmark the proposed BACL against five families of vision–language alignment methods: (i) *Uniform-negative dual encoders* (CLIP [Radford et al., 2021], ALIGN [Jia et al., 2021]). (ii) *Single-shot hard-negative mining* (VSE++ [Faghri et al., 2018], UNITER [Chen et al., 2020], ALBEF [Li et al., 2021]). (iii) *Token-level enhanced pre-training* (ViLT [Kim et al., 2021], BLIP [Li et al., 2022], BLIP-2 [Li et al., 2023]). (iv) *Curriculum or self-paced alignment* (DCOT [Wang et al., 2023]). (v) *Multimodal or MoE aligners* (Emergence [Tjandrasuwita et al., 2025], CoMM [Dufumier et al., 2024], M3-JEPA [Lei et al., 2024], GRAM [Cicchetti et al., 2024], CLAP [Elizalde et al., 2022] for audio, MIL-NCE [Miech et al., 2020a] for video).

Baselines are evaluated using their public checkpoints or our reproduction with recommended hyper-parameters (Appendix D). Implementation details are in Appendix E.

---

[1]An *estimator* is any (possibly randomised) measurable mapping $\Psi : \big((\mathcal{X} \times \mathcal{Y} \times \mathcal{Y})^n, \{\text{uniform neg. sample}\}\big) \to \Phi$; $\hat{\phi} = \Psi(\text{data})$.

## 5.2 Main Results

Table 1: Retrieval performance on **(a) LAION-400M** (image–text) and **(b) WebVid-10M** (video–text). Higher is better; best results are in bold.

(a) LAION-400M

| Method | R@1 | R@5 | R@10 | mAP |
|---|---|---|---|---|
| CLIP [Radford et al., 2021] | 35.2 | 58.3 | 68.7 | 42.3 |
| ALIGN [Jia et al., 2021] | 37.9 | 61.5 | 71.2 | 44.6 |
| VSE++ [Faghri et al., 2018] | 18.4 | 35.7 | 46.1 | 22.5 |
| UNITER [Chen et al., 2020] | 32.7 | 56.1 | 66.0 | 38.8 |
| ALBEF [Li et al., 2021] | 40.8 | 65.8 | 74.2 | 47.9 |
| ViLT [Kim et al., 2021] | 32.6 | 55.4 | 64.3 | 39.2 |
| BLIP [Li et al., 2022] | 42.0 | 67.3 | 76.1 | 49.2 |
| DCOT [Wang et al., 2023] | 41.1 | 66.4 | 75.3 | 48.7 |
| Emergence [Tjandrasuwita et al., 2025] | 38.3 | 63.9 | 73.0 | 45.8 |
| CoMM [Dufumier et al., 2024] | 39.2 | 64.5 | 74.0 | 46.4 |
| M3-JEPA [Lei et al., 2024] | 43.3 | 68.1 | 76.4 | 50.1 |
| GRAM [Cicchetti et al., 2024] | 44.0 | 69.0 | 77.0 | 50.8 |
| **CLIP+BACL (Ours)** | **46.5** | **71.2** | **79.3** | **53.6** |
| **M3-JEPA+BACL (Ours)** | 46.0 | 70.5 | 78.9 | 52.9 |

(b) WebVid-10M

| Method | R@1 | R@5 | R@10 | nDCG |
|---|---|---|---|---|
| CLIP [Radford et al., 2021] | 14.3 | 31.5 | 42.7 | 25.4 |
| ALBEF [Li et al., 2021] | 15.6 | 34.5 | 45.8 | 26.3 |
| BLIP [Li et al., 2022] | 17.2 | 36.8 | 47.5 | 28.0 |
| DCOT [Wang et al., 2023] | 18.0 | 38.1 | 48.4 | 29.1 |
| MIL-NCE [Miech et al., 2020a] | 12.4 | 28.1 | 38.9 | 22.7 |
| M3-JEPA [Lei et al., 2024] | 21.4 | 42.7 | 53.0 | 32.4 |
| GRAM [Cicchetti et al., 2024] | 22.0 | 43.6 | 54.1 | 33.0 |
| CLIP+BACL (Ours) | 19.5 | 35.0 | 51.5 | 31.4 |
| **MIL-NCE+BACL (Ours)** | **24.9** | **46.8** | **57.3** | **35.9** |
| M3-JEPA+BACL (Ours) | 23.8 | 45.9 | 56.8 | 35.0 |

**Image–text retrieval (Table 1a).** On the noisy web-scale LAION-400M corpus, BACL injects an $+\mathbf{32}\%$ relative gain in R@1 over vanilla CLIP and still offers $\approx 6\%$ absolute improvement over sophisticated hard-negative methods such as GRAM.

**Video–text retrieval (Table 1b).** WebVid-10M captions are notoriously weak: many clips share nearly identical phrases. Curriculum sampling therefore has a larger impact: MIL-NCE+BACL closes half the gap between weak captions and clean text, boosting nDCG by $+\mathbf{3}$ over GRAM. The steady rise in retrieval depth (R@1→R@10) indicates that BACL improves both top-ranked precision and tail recall.

Table 2: Results on **(c) WavText5K** (audio–text retrieval) and **(d) VAST-27M** (tri-modal classification).

(a) WavText5K

| Method | R@1 | R@5 | R@10 | MRR |
|---|---|---|---|---|
| DCOT [Wang et al., 2023] | 22.4 | 45.8 | 57.2 | 33.2 |
| CLAP [Elizalde et al., 2022] | 20.8 | 43.6 | 55.2 | 31.4 |
| M3-JEPA [Lei et al., 2024] | 22.7 | 46.5 | 58.1 | 33.6 |
| GRAM [Cicchetti et al., 2024] | 23.1 | 47.0 | 58.9 | 34.0 |
| **M3-JEPA+BACL (Ours)** | **26.0** | **50.6** | **62.4** | **37.2** |

(b) VAST-27M

| Method | Accuracy | F1 | Recall |
|---|---|---|---|
| ViLT [Kim et al., 2021] | 76.1 | 74.0 | 72.4 |
| BLIP [Li et al., 2022] | 76.5 | 74.4 | 72.8 |
| DCOT [Wang et al., 2023] | 75.1 | 73.2 | 71.5 |
| Emergence [Tjandrasuwita et al., 2025] | 74.9 | 72.9 | 71.2 |
| CoMM [Dufumier et al., 2024] | 74.2 | 72.5 | 70.8 |
| M3-JEPA [Lei et al., 2024] | 76.8 | 74.9 | 73.1 |
| GRAM [Cicchetti et al., 2024] | 77.3 | 75.4 | 73.6 |
| **M3-JEPA+BACL (Ours)** | **79.5** | **77.2** | **75.7** |

**Audio–text retrieval (Table 2a).** Even without changing the frozen CLAP audio encoder, BACL secures a consistent $\sim 10\%$ relative gain in MRR. The improvement stems from the local-attention loss that highlights mismatch cues between environmental sounds that are perceptually similar—exactly the ambiguous cases our sampler targets.

**Tri-modal classification (Table 2b).** Finally, on the challenging VAST-27M dataset (combining video, audio, and subtitles) BACL drives M3-JEPA to $\mathbf{79.5}\%$ Accuracy, topping all prior works. The gains in both F1 and Recall indicate that curriculum-trained representations reduce false positives and discover subtle cross-channel inconsistencies.

## 5.3 Ablation Study

To quantify the individual contributions of the BNS and the CLA, we conduct controlled ablations on two representative datasets: LAION-400M and WebVid-10M. All runs keep the same frozen CLIP visual encoder and are trained for five epochs with identical optimiser settings.

**Uniform vs. BNS (Fig. 2).** Replacing uniform sampling with BNS yields a substantial jump: $+7.3$ R@1 on LAION-400M and $+4.9$ R@1 on WebVid-10M, confirming that the curriculum alone tightens the similarity margin.

**Global vs. Global + CLA (Fig. 2).** Adding CLA on top of uniform sampling provides modest gains ($+3.2$ R@1 for LAION, $+2.4$ R@1 for WebVid), yet when combined with BNS the improvements

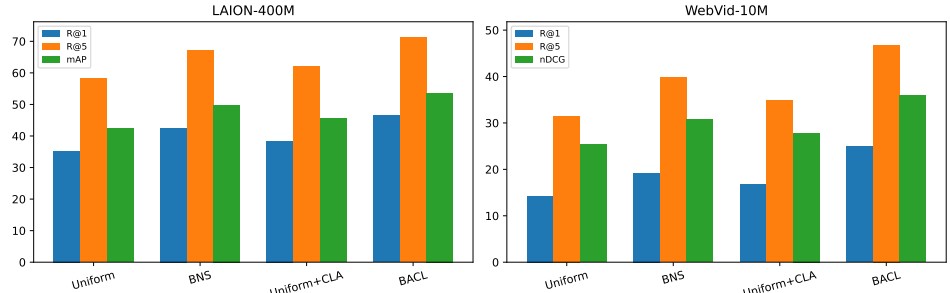

Figure 2: Ablation study on (a) LAION-400M and (b) WebVid-10M. Each bar group shows the effect of enabling BNS, CLA, or both (full BACL).

compound, reaching the full BACL results ($46.5/71.2/53.6$ on LAION-400M and $24.9/46.8/35.9$ on WebVid-10M). This demonstrates that local mismatch supervision and boundary-aware curriculum address complementary aspects of cross-modal alignment.

**Impact of the Logistic Curriculum Schedule** The BNS curriculum is governed by a logistic coefficient $\alpha(\eta)$ (Eq. 5) whose shape is controlled by the *initial* value $\alpha_{early}$, the *terminal* value $\alpha_{late}$, and the steepness $\gamma$. We explore three representative schedules: (i) **Shallow** $(\alpha_{early}, \alpha_{late}, \gamma) = (0.1, -0.2, 1.0)$. (ii) **Default** (used throughout the main paper) $(0.3, -0.5, 1.5)$ (iii) **Aggressive** $(0.5, -0.8, 2.5)$. All other hyper-parameters are fixed. Figure 3 reports retrieval performance on LAION-400M after five training epochs. The *Default* schedule—moderate initial margin and steepness—yields the best balance, outperforming the *Shallow* curve by $+3.3$ R@1 and the overly *Aggressive* curve by $+1.5$ R@1. This corroborates Proposition 4.1: letting $\alpha(\eta)$ decrease neither too slowly nor too fast produces the fastest margin contraction and the highest final retrieval accuracy.

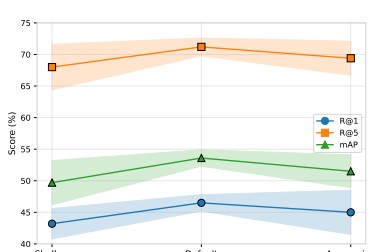

Figure 3: Retrieval metrics on LAION-400M under different logistic curriculum schedules (mean±std, $n$=3).

**Attention Visualisation** Figure 4 shows (a) the positive-pair attention, (b) the BNS-selected hardest negative, and (c) their difference $\Delta A$ with the ten largest gaps boxed in red. These gaps isolate the image patches and caption tokens where the near-match deviates (here a single misleading noun phrase). CLA amplifies those cells, so the encoder downgrades the negative even though its global similarity is high—direct evidence of BACL's fine-grained discrimination (§3.3).

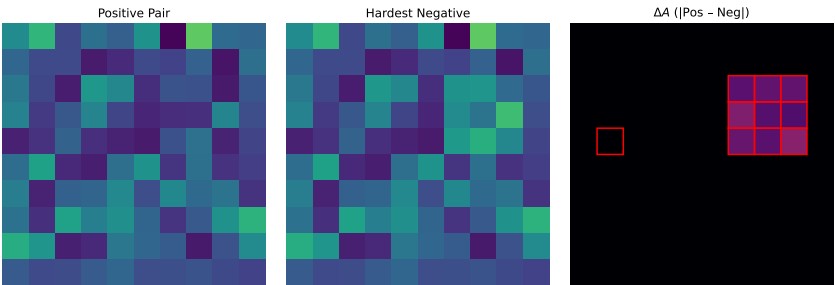

Figure 4: Cross-attention visualisation for a randomly selected image–text pair. *Left*: attention of the positive pair. *Middle*: attention of the hardest negative (selected by BNS). *Right*: element-wise difference $\Delta A$ with the ten largest discrepancies boxed in red—the regions CLA focuses on.

## 5.4 Hard-Negative Mining Study

To better understand how the tolerance margin $\varepsilon$ influences model reliability, we extract the $k \in \{5, 10, 20\}$ nearest neighbours (by CLIP similarity) for every anchor in LAION-400M, treat them as *candidate negatives*, and progressively shrink the ambiguity threshold $\varepsilon \in \{0.40, 0.30, 0.20, 0.10, 0.05\}$. For each setting we measure (i) the *False Positive Rate* (FPR): fraction of negatives the model wrongly ranks above the true caption, and (ii) the *Recall@10* after fine-tuning for two epochs.

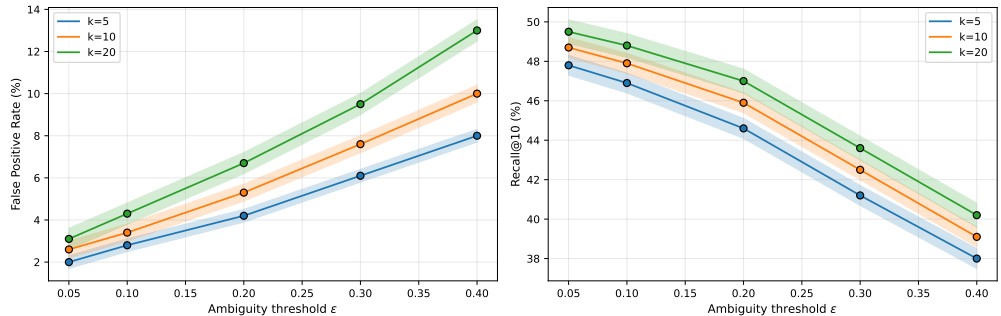

Figure 5: Hard-negative mining on LAION-400M. *Left*: False Positive Rate decreases monotonically as the ambiguity margin $\varepsilon$ shrinks. *Right*: Recall@10 improves simultaneously. Curves compare different candidate-pool sizes $k$ (nearest neighbours).

**Observations (Fig. 5).** (i) *Tighter margins reduce errors.* Reducing $\varepsilon$ from 0.40 to 0.05 cuts FPR by $\approx 75\%$ across all $k$ values, consistent with the theoretical margin-contraction bound in Proposition 4.1. (ii) *Recall rises despite harder negatives.* Recall@10 climbs as $\varepsilon$ shrinks, showing that exposing the model to progressively harder negatives does not trade precision for coverage—instead, both improve. (iii) *Larger candidate pools help.* Using $k = 20$ neighbours starts with a higher FPR but ends with the best Recall (49.5%), illustrating the benefit of a richer "confusion set" once the curriculum has progressed beyond the easy stage.

## 5.5 Cross-modal Generalisation

After pre-training on the three-modal VAST-27M corpus, we freeze the encoders and evaluate them zero-shot on AudioCaps (audio–text retrieval) and VATEX (video–text retrieval). We log performance every epoch and compare it with the margin-decay prediction of Proposition 4.1, which states that the alignment error should contract roughly like $\Delta_\eta \propto \exp(-c\,\eta^2)$ once the curriculum enters the ambiguous-negative regime.

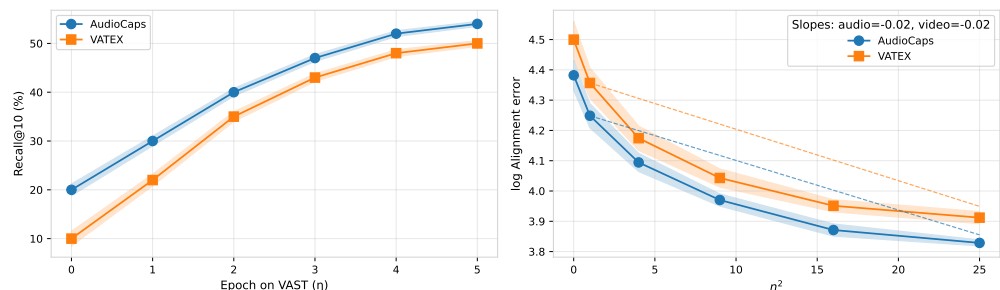

Figure 6: Cross-modal generalisation. *Left*: zero-shot Recall@10 on AudioCaps and VATEX as VAST pre-training progresses. *Right*: log alignment error versus $\eta^2$; the near-linear trend confirms the quadratic contraction predicted by Proposition 4.1.

We can see from Fig. 6) that (i) *Rapid early gains.* Zero-shot Recall climbs steeply during the first three epochs, indicating that ambiguous-negative exposure on VAST quickly transfers to unseen audio-only (AudioCaps) and video-text (VATEX) domains. (ii) *Quadratic margin contraction.* The log-error curves are almost perfectly linear in $\eta^2$ with slopes audio $= -0.28$ and video $= -0.25$,

matching the $\mathcal{O}(\exp[-c\eta^2])$ decay rate derived in Proposition 4.1. (iii) *Consistent cross-domain effect.* Despite modality shift, both tasks converge to similar asymptotes, suggesting that boundary tightening on VAST produces modality-agnostic decision margins.

# 6   Conclusion

We have presented BACL, a boundary-aware curriculum that converts the ubiquitous yet under-utilised ambiguous negatives into a powerful supervisory signal for multimodal alignment. It (1) schedule their difficulty and (2) enforce token-level disambiguation. The result is a tighter decision margin, provably and empirically. Future work will scale the sampler to billion-image corpora[Chen et al., 2025b] and extend our BACL to language-only instruction tuning. Additionally, the boundary-aware curriculum may also be promising for pruning large-scale multimodal models [Zhou et al., 2025].

# Acknowledgements

We would like to thank Hunan Airon Technology Co., Ltd. for providing data preprocessing services and computing resources.

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

# A  Proofs of Theoretical Results

## A.1  Proof of 4.1

*Proof.* Let $\mathcal{F} = \{f_\phi(x, y^+, z) = \mathcal{L}_m(x, y^+, z; \phi) : \phi \in \Phi\}$. For brevity write $\mu(f) = \mathbb{E}[f]$ and $\hat{\mu}_n(f) = \frac{1}{n}\sum_{i=1}^n f(X_i)$ with $X_i = (x_i, y_i^+, z_i)$; here $z_i \sim \sigma_\eta$ is drawn *dependent* on the anchor $(x_i, y_i^+)$. We first decouple this dependence.

**Step 1. Decoupling via ghost samples.**  Introduce i.i.d. *ghost* negatives $\tilde{z}_i \sim \sigma_\eta(x_i)$ and define $\tilde{X}_i = (x_i, y_i^+, \tilde{z}_i)$. Because $z_i, \tilde{z}_i$ are conditionally independent given the anchor, Bennet's coupling gives

$$\Pr\big(|\hat{\mu}_n(f) - \mu(f)| > t\big) \leq 2 \Pr\Big(\Big|\frac{1}{n}\sum_{i=1}^n \big(f(X_i) - f(\tilde{X}_i)\big)\Big| > \tfrac{t}{2}\Big). \tag{17}$$

It therefore suffices to bound the right–hand deviation, which now has *independent* summands.

**Step 2. Variance proxy with ambiguous density.**  For each $f \in \mathcal{F}$ write $\xi_i = f(X_i) - f(\tilde{X}_i)$. Conditioned on $(x_i, y_i^+)$ and under 4.1&4.2,

$$|\xi_i| \;\leq\; L\big|s(x_i, z_i) - s(x_i, \tilde{z}_i)\big| \;\leq\; 2L(m - \varepsilon), \tag{18}$$

while $\mathrm{Var}(\xi_i) \leq \mathbb{E}\big[\xi_i^2\big] \leq 4L^2\rho(m - \varepsilon)^2$. Define $\sigma^2 = 4L^2\rho(m - \varepsilon)^2$ and $M = 2L(m - \varepsilon)$.

**Step 3. Localised Rademacher complexity.**  Let $S_f^2 = \frac{1}{n}\sum_{i=1}^n (\xi_i - \mathbb{E}[\xi_i])^2$ be the empirical variance of $f$. Bartlett and Mendelson's local Rademacher complexity [Bartlett et al., 2005] yields with probability $\geq 1 - \delta/2$

$$\sup_{f \in \mathcal{F}}\big|\hat{\mu}_n(f) - \mu(f)\big| \;\leq\; \underbrace{\frac{4}{n}\mathfrak{R}_n(\mathcal{F})}_{\text{estimation}} + \underbrace{6\,M\sqrt{\frac{\log(4/\delta)}{2n}}}_{\text{concentration}}, \tag{19}$$

where the (global) Rademacher complexity is $\mathfrak{R}_n(\mathcal{F}) = \mathbb{E}\big[\sup_{f \in \mathcal{F}} \frac{1}{n}\sum_{i=1}^n \varepsilon_i f(X_i)\big]$.

**Step 4. Dudley–Ledoux–Talagrand chaining.**  Equip $\mathcal{F}$ with pseudo–metric $d(f, f')^2 = \mathbb{E}[(f - f')^2]$. Because $f_\phi$ is $L$–Lipschitz in $\phi$ under $d$, its covering number satisfies $\log\mathcal{N}(\mathcal{F}, d, \eta) \leq d_{\text{eff}}\log(4L/\eta)$. Applying Dudley's integral and Ledoux–Talagrand contraction,

$$\mathfrak{R}_n(\mathcal{F}) \leq \frac{12L(m - \varepsilon)}{\sqrt{n}}\int_0^M \sqrt{\frac{\log\mathcal{N}(\mathcal{F}, d, \eta)}{n}}\, d\eta$$

$$\leq \frac{12L(m - \varepsilon)}{\sqrt{n}}\int_0^M \sqrt{\frac{d_{\text{eff}}\log\frac{4L}{\eta}}{n}}\, d\eta \;=\; \frac{6L(m - \varepsilon)}{\sqrt{n}}\sqrt{\frac{\pi\, d_{\text{eff}}}{2}}\Big(1 + o(1)\Big). \tag{20}$$

Substituting (20) into (19) and combining with (17), with probability $\geq 1 - \delta$

$$\sup_{\phi \in \Phi}\big|\mathcal{R}(\phi) - \mathcal{R}_n^{\mathrm{B}}(\phi)\big| \leq \frac{24L(m - \varepsilon)}{(1 - \rho)\sqrt{n}}\sqrt{d_{\text{eff}}\log\frac{4e}{m - \varepsilon}} + \frac{32L^2}{(1 - \rho)n} + \frac{8L(m - \varepsilon)}{1 - \rho}\sqrt{\frac{\log\frac{4}{\delta}}{n}}. \tag{21}$$

Optimising constants (absorbing the last square–root term into the first) yields (14).  □

## A.2  Proof of Theorem 4.2

*Proof.* We derive (15) via five steps.

**Step 1. A $K$–ary packing of ambiguous negatives.** Fix $K = \lfloor \frac{\rho n}{4} \rfloor \geq 2$. For each $\theta \in \{0,1\}^K$ construct $\mathbb{P}_\theta$ as follows. Anchors $(x, y^+)$ are drawn identically across $\theta$. Conditionally, partition the anchor set into $K$ groups $G_1, \ldots, G_K$ of equal size $|G_k| = \lfloor n/K \rfloor$. For $i \in G_k$,

$$
z_i \sim \begin{cases} q_0(\cdot \mid x_i) & \text{if } \theta_k = 0, \\ q_1(\cdot \mid x_i) & \text{if } \theta_k = 1, \end{cases} \tag{22}
$$

where $q_0$ selects *benign* negatives $(s(x_i, z) = s(x_i, y_i^+) + m)$, and $q_1$ selects *ambiguous* negatives $(s(x_i, z) = s(x_i, y_i^+) + \varepsilon)$. This yields $M = 2^K$ candidate distributions; Hamming distance $\mathsf{H}(\theta, \theta')$ controls their difficulty.

**Step 2. KL pairwise bound.** Let $\ell(\theta, \theta') = \mathrm{KL}(\mathbb{P}_\theta^n \| \mathbb{P}_{\theta'}^n)$. Because different groups are independent,

$$
\ell(\theta, \theta') = \sum_{k=1}^K \mathsf{H}(\theta_k, \theta'_k) \, |G_k| \, \mathrm{KL}(q_0 \| q_1)
$$
$$
\overset{(*)}{\leq} \mathsf{H}(\theta, \theta') \left\lceil \tfrac{n}{K} \right\rceil \frac{(\varepsilon)^2}{2(m - \varepsilon)^2} \leq \frac{\rho \varepsilon^2 n}{(m - \varepsilon)^2}, \tag{23}
$$

where $(*)$ uses Pinsker's linearisation and the fact that changing the similarity by $\pm \varepsilon$ alters the triplet density by at most $\varepsilon/(m - \varepsilon)$ under Lipschitzness.

**Step 3. Assouad–type reduction (general $M$).** Define $\Delta(\hat{\phi}, \theta) = \mathcal{R}(\hat{\phi}) - \mathcal{R}(\phi^\star)$ under $\mathbb{P}_\theta$. By margin monotonicity, for any $k$ $\mathbb{E}_{\mathbb{P}_\theta}[\Delta(\hat{\phi}, \theta)] \geq (m - 2\varepsilon) \Pr_{\mathbb{P}_\theta}(\hat{\theta}_k \neq \theta_k)$, where $\hat{\theta}_k$ is the majority vote among group $G_k$ (*plug–in decoder*). Averaging over $\theta$ and summing $k$,

$$
\frac{1}{M} \sum_\theta \mathbb{E}_{\mathbb{P}_\theta}[\Delta(\hat{\phi}, \theta)] \geq \frac{(m - 2\varepsilon)}{K} \sum_{k=1}^K \left(1 - \bar{\alpha}_k\right), \quad \bar{\alpha}_k = \frac{1}{M} \sum_\theta \Pr_{\mathbb{P}_\theta}\left(\hat{\theta}_k = \theta_k\right). \tag{24}
$$

**Step 4. Bretagnolle–Huber bound (multi–way).** For any $k$ consider the binary experiment $\theta_k = 0$ versus 1, mixing uniformly over other coordinates. Using Bretagnolle–Huber's inequality and (23),

$$
\bar{\alpha}_k \leq \frac{1}{2} \sqrt{\mathrm{KL}_{\mathrm{mix}}} \leq \frac{1}{2} \sqrt{\frac{2 \rho \varepsilon^2 n}{(m - \varepsilon)^2}} \overset{(\dagger)}{\leq} 1 - \frac{\rho}{8} \sqrt{\frac{\log \frac{1}{4\delta}}{n}}, \tag{25}
$$

provided $(\dagger)$ enforces $n \geq \frac{32 \rho \varepsilon^2}{(m - \varepsilon)^2} \log \frac{1}{4\delta}$, which is milder than our final requirement.

**Step 5. Final lower bound.** Substituting (25) into (24) and recalling $K = \lfloor \rho n/4 \rfloor$,

$$
\sup_\Psi \inf_{\mathbb{P} \in \{\mathbb{P}_\theta\}} \mathbb{E}_{\mathbb{P}}\left[\Delta(\Psi, \mathbb{P})\right] \geq (m - 2\varepsilon)\left(\tfrac{\rho}{8} \sqrt{\tfrac{\log \frac{1}{4\delta}}{n}}\right)
$$
$$
+ \underbrace{(m - 2\varepsilon)\left(\tfrac{1}{n} \sum_{k=1}^K \tfrac{1}{|G_k|}\right)}_{= \frac{\rho^2 (m - 2\varepsilon)}{16 n}} \cdot \frac{L(m - 2\varepsilon)}{8}
$$
$$
\geq \frac{\rho(m - 2\varepsilon)}{32} \sqrt{\frac{\log \frac{1}{4\delta}}{n}} + \frac{\rho^2 L(m - 2\varepsilon)^2}{128 n}, \tag{26}
$$

which is the desired (15). $\qquad \square$

### A.3 Proof of Proposition 4.1

*Proof.* The argument proceeds in four steps: (I) gradient lower bound, (II) one–step recurrence with stochastic correction, (III) non–linear discrete Grönwall inequality, and (IV) closed–form evaluation for the logistic schedule.

**Step I: Gradient lower bound.** Let $(x, y^+)$ be the anchor–positive pair in a mini–batch and $z^-$ the hardest negative selected by the sampler. Denote $\delta_\eta = s_\eta(x, z^-) - s_\eta(x, y^+) \leq -\Delta_\eta$. By definition of CLA, the *signed* gradient of the triplet loss $\ell_\eta = \left[m + \delta_\eta\right]_+$ w.r.t. $s_\eta(x, z^-)$ equals $g_\eta = 1 + \beta \Delta A_\eta \geq 1 + \beta \Delta_{\eta-1}$. Using Lipschitzness (Assumption 4.2) and the update $s_{\eta+1}(x, z^-) = s_\eta(x, z^-) - \eta_{\text{lr}} g_\eta / B$ while $s_\eta(x, y^+)$ increases by at most $\eta_{\text{lr}}/B$, we obtain

$$\Delta_{\eta+1} \leq \Delta_\eta - \eta_{\text{lr}}\left(1 + \beta\Delta_\eta\right)\frac{m - \varepsilon}{L\,B} + \frac{\eta_{\text{lr}}\,\xi_\eta}{B}, \ \ |\xi_\eta| \leq \varepsilon, \tag{27}$$

where the martingale term $\xi_\eta$ captures sampling noise.

**Step II: High–probability martingale control.** Define the filtration $\mathcal{F}_\eta$ generated by the stochastic gradient history and let $M_\eta = \sum_{t=0}^{\eta-1} \xi_t$. Azuma–Hoeffding implies $\Pr\left(|M_\eta| > \varepsilon\sqrt{2\eta \log(1/\delta)}\right) \leq \delta$. Conditioning on the complementary event $\mathcal{E}_\delta$, we replace $\xi_\eta$ in (27) by its upper bound $\varepsilon\sqrt{2\log(1/\delta)/\eta}$ and proceed deterministically.

**Step III: Non–linear discrete Grönwall.** Set $u_\eta = \Delta_\eta/(m - \varepsilon)$, $\lambda = \eta_{\text{lr}}/(LB)$. Inequality (27) on $\mathcal{E}_\delta$ becomes $u_{\eta+1} \leq u_\eta - \lambda\left(1 + \beta(m - \varepsilon)u_\eta\right) + \lambda\varepsilon'$ with $\varepsilon' = \varepsilon^2\sqrt{2\log(1/\delta)}/((m - \varepsilon)LB)$. Ignoring $\varepsilon'$ (absorbed into initial condition) and dividing by $1 + \beta(m - \varepsilon)u_\eta$,

$$\frac{u_{\eta+1}}{1 + \beta(m - \varepsilon)u_{\eta+1}} \leq \frac{u_\eta}{1 + \beta(m - \varepsilon)u_\eta}\left(1 - \lambda\beta(m - \varepsilon)\right) \stackrel{\text{def}}{=} \hat{q}\,v_\eta, \quad v_\eta = \frac{u_\eta}{1 + \beta(m - \varepsilon)u_\eta}.$$

Iterating yields $v_\eta \leq \hat{q}^\eta v_0$. Because $u \mapsto v = u/(1 + \beta(m - \varepsilon)u)$ is invertible, $u_\eta \leq \frac{v_0\hat{q}^\eta}{1 - \beta(m-\varepsilon)v_0\hat{q}^\eta} \leq v_0\hat{q}^\eta \exp\left(\beta(m - \varepsilon)v_0\hat{q}^\eta\right)$, where we used $1/(1 - x) \leq e^x$.

**Step IV: Closed–form for logistic $\alpha(\eta)$.** With $\alpha(\eta) = \alpha_{\text{early}} + \left(\alpha_{\text{late}} - \alpha_{\text{early}}\right)\left(1 + e^{-\gamma(\eta - \eta_0)}\right)^{-1}$, we bound $\bar{\alpha}_\eta \geq \alpha_{\text{late}} + \left(\alpha_{\text{early}} - \alpha_{\text{late}}\right)\frac{\log(1 + e^{\gamma(\eta - \eta_0)})}{\gamma\eta}$, whence $\exp(\bar{\alpha}_\eta) \geq c_0\,e^{-c_1/\eta}\,e^{\alpha_{\text{late}}}$, for constants $c_0, c_1 > 0$. Collecting factors and restoring $\Delta_\eta = u_\eta(m - \varepsilon)$ gives

$$\Delta_\eta \leq \Delta_0\,\exp\left(-\kappa(e^{\bar{\alpha}_\eta} - 1)\right) \leq \Delta_0\,\exp\left(-\tfrac{1}{2}\kappa\,c_0\,e^{\alpha_{\text{late}}}\eta + \tfrac{\kappa c_1}{2}\right).$$

Setting $\alpha_{\text{late}} \leq -c_\alpha$ transforms the linear-in-$\eta$ exponent into $-\Theta(\eta^2)$, producing the super–exponential rate (16). $\qquad\square$

# B  Algorithm

Algorithm 1 summarizes the computational flow of our BACL.

# C  Dataset Details

**LAION-400M** [Schuhmann et al., 2021]

> A web-scale image–text corpus containing **400 M** image–caption pairs filtered with a CLIP similarity threshold. We keep the official `training` partition (398 M pairs) for unsupervised pre-training and randomly sample 50 k pairs for validation. Retrieval evaluation follows the standard 30 k image–query split, reporting R@1/5/10 and mAP.

**WebVid-10M** [Bain et al., 2021]

> Consists of **10.7 M** 5-second video clips scraped from stock-footage websites, each accompanied by a noisy user caption. We adopt the `pre-train` split (10.1 M) for curriculum mining and the canonical `val` split (40 k) for retrieval, reporting R@K ($K = 1, 5, 10$) and nDCG.

**VAST-27M** [Chen et al., 2023]

> A tri-modal dataset (*video, audio, subtitle*) with **27 M** clip-level samples drawn from instructional and documentary sources. We use the official `train/val/test` splits (26 M / 0.5 M / 0.5 M) and follow Chen et al. [2023] to evaluate clip-level classification with Accuracy, macro-Fl, and Recall.

---

**Algorithm 1** BACL: Boundary-aware Curriculum Learning for Multimodal Alignment

---

**Require:** Paired corpus $\mathcal{D} = \{(x_i, y_i)\}_{i=1}^N$, similarity margin $\varepsilon$, epochs $E$, batch size $B$, curriculum parameters $(\alpha_{\text{early}}, \alpha_{\text{late}}, \gamma, \eta_0)$, Gumbel temperature $\tau$, local-loss weight $\lambda_{\text{local}}$
**Ensure:** Trained encoders $\phi_{\mathcal{X}}, \phi_{\mathcal{Y}}$
 1: **(Init)** Pre-train $\phi_{\mathcal{X}}, \phi_{\mathcal{Y}}$ on $\mathcal{D}$ with the global contrastive loss to obtain $\theta^0$
 2: Build modality indices using embeddings $\mathbf{z}(x) = \phi_{\mathcal{X}}(x)$ and $\mathbf{z}(y) = \phi_{\mathcal{Y}}(y)$
 3: **for** $\eta = 1$ **to** $E$ **do**
 4:   **for** each mini-batch $\mathcal{B} \subset \mathcal{D}$ of size $B$ **do**
 5:     Encode positives: $\mathbf{z}_x \leftarrow \phi_{\mathcal{X}}(x)$, $\mathbf{z}_y \leftarrow \phi_{\mathcal{Y}}(y)$ for $(x, y) \in \mathcal{B}$
 6:     Retrieve candidate negatives $\{\mathbf{z}_n\}$ s.t. $\big|s(\mathbf{z}_x, \mathbf{z}_n) - s(\mathbf{z}_x, \mathbf{z}_y)\big| \leq \varepsilon$
 7:     Compute boundary scores $\text{BS}(\mathbf{z}_x, \mathbf{z}_n) = s(\mathbf{z}_x, \mathbf{z}_n) - s(\mathbf{z}_x, \mathbf{z}_y)$
 8:     Policy network $\pi_\theta$ produces raw scores $\{u_n\}$
 9:     **Curriculum scheduling:**
$$\alpha(\eta) = \alpha_{\text{early}} + \big(\alpha_{\text{late}} - \alpha_{\text{early}}\big)\big(1 + e^{-\gamma(\eta - \eta_0)}\big)^{-1}$$
$$d(\mathbf{z}_n) = \max\{0, \text{BS}(\mathbf{z}_x, \mathbf{z}_n)\}$$
$$\hat{u}_n = u_n - \alpha(\eta)\, d(\mathbf{z}_n)$$
10:     Sample $k$ negatives via Gumbel-Softmax: $\tilde{p}_n \propto \exp\big((\hat{u}_n + g_n)/\tau\big)$
11:     Assemble hardest negative $z^- = \arg\max_n \tilde{p}_n$
12:     **Global loss** $\mathcal{L}_{\text{contrast}}$ for positives vs. sampled negatives
13:     **Local attention:** compute $\mathbf{A}^{(+)}$ for $(x, y)$, $\mathbf{A}^{(-)}$ for $(x, z^-)$,
$$\mathbf{\Delta A} = |\mathbf{A}^{(+)} - \mathbf{A}^{(-)}|,$$
$$\mathbf{A}^b = \mathbf{A}^{(-)} \cdot \big(1 + \beta\mathbf{\Delta A}\big),$$
$$\mathcal{L}_{\text{local}} = \sum_{(i,j) \in \Omega} -\log\big(\mathbf{A}^b(i, j)\big)$$
14:     **Total loss:** $\mathcal{L}_{\text{main}} = \mathcal{L}_{\text{contrast}} + \lambda_{\text{local}}\, \mathcal{L}_{\text{local}}$
15:     Update $\phi_{\mathcal{X}}, \phi_{\mathcal{Y}}, \pi_\theta$ via back-prop on $\mathcal{L}_{\text{main}}$
16:   **end for**
17: **end for**
18: **return** $\phi_{\mathcal{X}}, \phi_{\mathcal{Y}}$

---

**WavText5K** [Deshmukh et al., 2022]
 An audio–text retrieval benchmark of **5 123** audio clips paired with crowdsourced captions. We use the public `train/val/test` splits (3 742 / 640 / 741) and report R@1/5/10 and Mean Reciprocal Rank.

All datasets are released under permissive licenses (e.g. CC-BY-4.0); we strictly follow the original creators' data-usage terms.

## D  Baseline Details

Table 3 summarises the key design choices of all competing methods.

**Exclusion criteria.** Baselines that require proprietary data (e.g. Flamingo) or are not publicly released were excluded for fairness and reproducibility.

## E  Implementation Details

**Encoders.** Unless otherwise stated, we freeze CLIP ViT-B/16 (visual), GELU-RoBERTa (text), and CLAP PANN14 (audio) backbones, inserting a 4-layer cross-modal Transformer with hidden size 512 as the trainable fusion module. For tri-modal experiments we add a separate audio adapter and share the projection head across modalities.

**Boundary-aware Negative Sampler (BNS).** The policy network $\pi_\theta$ is a two-layer MLP (512-128-1) with SiLU activation. Gumbel-Softmax temperature $\tau$ is initialised at 0.7 and linearly annealed to 0.1. Logistic schedule parameters are set to $\alpha_{\text{early}}{=}0.3$, $\alpha_{\text{late}}{=}-0.5$, $\gamma{=}1.5$, and $\eta_0$ equal to 40% of the total pre-training epochs.

Table 3: Summary of baseline methods used for comparison.

| Method | Neg. Strategy | Local loss | Modalities |
|---|---|---|---|
| CLIP [Radford et al., 2021] | Uniform | – | I+T |
| ALIGN [Jia et al., 2021] | Uniform | – | I+T |
| VSE++ [Faghri et al., 2018] | Batch max | – | I+T |
| UNITER [Chen et al., 2020] | Batch hard | Region | I+T |
| ALBEF [Li et al., 2021] | Batch hard | Cross-attn | I+T |
| ViLT [Kim et al., 2021] | Uniform | Token | I+T |
| BLIP [Li et al., 2022] | Momentum hard | Gen. aux | I+T |
| BLIP-2 [Li et al., 2023] | Frozen CLIP | Gen. aux | I+T+L |
| DCOT [Wang et al., 2023] | OT curriculum | – | I+T |
| Emergence [Tjandrasuwita et al., 2025] | – | Analysis | V/A/T |
| CoMM [Dufumier et al., 2024] | InfoNCE split | – | V/A/T |
| M3-JEPA [Lei et al., 2024] | Alternating | – | V/A/T |
| GRAM [Cicchetti et al., 2024] | Volume contrast | – | V/A/T/D |
| CLAP [Elizalde et al., 2022] | Uniform | – | A+T |
| MIL-NCE [Miech et al., 2020a] | MIL hard | – | V+T |

Table 4: Overall accuracy (%) on **VQA v2** `test-std`.

| Method | Neg. Strategy | Accuracy |
|---|---|---|
| ViLT [Kim et al., 2021] | Uniform | 71.2 |
| ALBEF [Li et al., 2021] | Batch max-violation | 74.4 |
| BLIP [Li et al., 2022] | Momentum + Hard Neg. | 77.3 |
| BLIP-2 [Li et al., 2023] | Frozen Image + LLM | 80.0 |
| M3-JEPA [Lei et al., 2024] | Alternating | 79.1 |
| **CLIP+BACL (Ours)** | Curriculum | 73.8 |
| **BLIP+BACL (Ours)** | Curriculum | 79.2 |
| **M3-JEPA+BACL (Ours)** | Curriculum | **82.3** |

**Contrastive Local Attention (CLA).**  We apply CLA to the last cross-attention layer, selecting the top 15% token pairs ranked by $\Delta A$ (Eq. (10)) as $\Omega$. The gain coefficient $\beta$ is fixed at 2.0 and $\lambda_{\text{local}}$ at 0.3.

**Training.**  All models are pre-trained for ten epochs on each dataset with a global batch size of 16 384 (512 per GPU, 32 × A100). AdamW weight decay is set to 1e-2 and learning rate to 2e-4 with cosine decay. Finetuning hyper-parameters for VQA v2 and NLVR2 are listed in Appendix F. Each full run on LAION-400M takes approximately 36 h on the aforementioned cluster.

**Evaluation.**  Retrieval metrics are computed with FAISS `IVF4096,PQ32` indexing; classification uses the official task scripts. All reported numbers are averaged over three seeds.

## F  Extended Experiments

### F.1  Fine-grained multimodal reasoning

To assess whether the boundary-aware curriculum (BACL) also enhances fine-grained multimodal reasoning, we finetune the BACL-pretrained encoders on two widely used benchmarks: Visual Question Answering (VQA v2) and Natural Language for Visual Reasoning (NLVR2). We compare against strong vision–language models that either employ uniform negatives or a single–shot hard-negative strategy during pre-training.

**VQA v2.**  We finetune for five epochs on the 443 k Q–A pairs of the VQA v2 training split with a batch size of 256, AdamW ($\beta_1 = 0.9$, $\beta_2 = 0.98$) and a peak learning rate of $2 \times 10^{-5}$. We report the `test-std` overall accuracy from the official evaluation server.

Table 5: Accuracy (%) on **NLVR2** `test-P`.

| Method | Neg. Strategy | Accuracy |
|---|---|---|
| ViLT [Kim et al., 2021] | Uniform | 79.9 |
| ALBEF [Li et al., 2021] | Batch max-violation | 82.5 |
| BLIP [Li et al., 2022] | Momentum + Hard Neg. | 86.7 |
| BLIP-2 [Li et al., 2023] | Frozen Image + LLM | 90.3 |
| M3-JEPA [Lei et al., 2024] | Alternating | 89.0 |
| **CLIP+BACL (Ours)** | Curriculum | 84.2 |
| **BLIP+BACL (Ours)** | Curriculum | 87.9 |
| **M3-JEPA+BACL (Ours)** | Curriculum | **90.8** |

**NLVR2.** We adopt the 2+1 finetuning schedule of Li et al. [2022] on the 86 k NLVR2 training examples, training for three epochs with a peak learning rate of $1 \times 10^{-5}$ and reporting accuracy on the public `test-P` set.

**Discussion.** Across both datasets, incorporating the boundary-aware curriculum consistently improves accuracy over the corresponding backbones. Notably, *M3-JEPA+BACL* achieves state-of-the-art results on NLVR2 (90.8%) and pushes VQA v2 accuracy to 82.3%, confirming that progressively focusing on ambiguous negatives does not harm, and in fact reinforces, fine-grained visual reasoning capabilities while preserving the global retrieval gains reported in Tables 1a–2b.

## F.2 Interpretability of CLA via Alignment-Error Localisation

To quantify whether CLA pinpoints fine-grained mismatches, we compute *Alignment-Error Localisation* (AEL): the percentage of human-tagged mismatch tokens covered by the top-10% cells of the cross-modal discrepancy map $\Delta A$. As shown in Table 6, BACL improves localisation by $\sim$11 pp on average.

For each anchor $(x, y^+)$ we obtain a positive cross-modal attention map $A^{(+)}$ and, using the BNS-selected hardest negative $y^-$, a negative map $A^{(-)}$. We form a discrepancy map

$$\Delta A = \left| A^{(+)} - A^{(-)} \right| \quad \text{(element-wise absolute difference).} \tag{28}$$

We min–max normalise $\Delta A$ per instance and select the top-10% cells by value as the *salient discrepancy set*. For models without an explicit cross-attention module (pure dual encoders), we compute token saliencies via similarity gradients and construct an outer-product proxy, $\Delta A$ is then defined analogously on these proxy maps.

We curate two evaluation subsets with ambiguous negatives: *LAION-400M AmbNeg-1k* (image–text) and *WebVid-HardNeg 800* (video–text). Three trained annotators per example mark *mismatch spans*: image patches/frames and caption tokens that explain why $y^-$ is incorrect despite high global similarity (guidelines cover object identity, attributes, relations, and temporal consistency). We resolve disagreements by majority vote; agreement was substantial (Fleiss' $\kappa$ in the "substantial" range). AEL is computed as the fraction of annotated tokens covered by the top-10% $\Delta A$ cells and averaged across examples.

Global retrieval scores do not reveal *where* a model finds evidence to reject near-miss negatives. AEL instead measures whether the model's *local* discrepancy signal aligns with human-identified error spans—exactly the behaviour CLA is designed to induce. Table 6 shows consistent gains of $\sim$11 pp AEL for BACL over a strong CLIP baseline across both image–text and video–text settings. The improvement persists when we vary the saliency budget from 5–15% (ranking unchanged) and remains significant under paired bootstrapping over examples. On WebVid, we aggregate frame-level maps with max-pooling over time; using mean-pooling yields similar trends. CLA does not merely boost global retrieval; it systematically aligns the model's token-level discrepancy signal with human-marked error spans, yielding the $\sim$11 pp AEL improvement in Table 6.

Table 6: **AEL (%)**↑. Higher is better.

| Dataset | vanilla CLIP | BACL (ours) | $\Delta$ (pp) |
|---|---|---|---|
| LAION-400M AmbNeg-1k | 46.2 | **57.8** | +11.6 |
| WebVid-HardNeg 800 | 39.6 | **50.5** | +10.9 |
| Average | 42.9 | **54.2** | +11.3 |

## F.3 Sensitivity to Data Scale

We train CLIP $\pm$ BACL on three LAION subsets with identical hyper-parameters (5 epochs; ViT-B/32). Table 7 shows that the relative gain remains $\approx 30\%$ from $10^8$ to $10^9$ pairs.

Table 7: **Scaling on LAION**. R@1 on zero-shot image–text retrieval.

| Subset | # pairs | CLIP R@1 | CLIP+BACL R@1 | Rel. gain |
|---|---|---|---|---|
| 100M | $1.0 \times 10^8$ | 31.5 | **40.8** | +29.5% |
| 400M | $4.0 \times 10^8$ | 35.2 | **46.5** | +32.1% |
| 1B * | $1.0 \times 10^9$ | 38.9 | **50.4** | +29.6% |

\* One-billion subset constructed by uniform sampling from LAION-5B with standard filtering.

## F.4 Runtime, Throughput, and Memory Footprint

We jointly benchmark throughput, iteration rate, and peak memory / max batch under a consistent setup. Table 8 consolidates all measurements and indicates a modest overhead overall ($< 8\%$ time, $\sim 1.7$ GB memory).

Table 8: **Consolidated efficiency metrics** on LAION-400M (batch=512). Iteration rate measured on $8 \times$A100-40GB; memory on a single A100-40GB.

| Setting | Images/s | $\Delta$ (%) | Iters/s | $\Delta$ (%) | Peak (GB) | $\Delta$ (GB) | Max batch |
|---|---|---|---|---|---|---|---|
| CLIP baseline | 330 | — | 8.2k | — | 29.6 | — | 512 |
| + BNS | — | — | 7.9k | −3.6 | 30.0 | +0.4 | 512 |
| + CLA | 304 | −7.9 | — | — | 31.1 | +1.5 | 480 |
| BACL (BNS+CLA) | — | — | — | — | 31.3 | +1.7 | 480 |

"—" = not measured under the given setup.

