# OpenReview forum: "Aligning by Misaligning: Boundary-aware Curriculum Learning for Multimodal Alignment"
_NeurIPS.cc/2025/Conference — NeurIPS 2025 poster_

### Official Review · Reviewer_Rymo · 2025-07-02

**Clarity:** 3
**Significance:** 3
**Originality:** 3
**Rating:** 4
**Confidence:** 4

**Summary:**

This paper introduces Boundary-Aware Curriculum with Local Attention (BACL), a novel framework for improving multimodal alignment by leveraging ambiguous negatives—pairs that are near the decision boundary and difficult to classify. ​ BACL consists of two key components:

Boundary-aware Negative Sampler (BNS): Dynamically schedules training samples from easy to hard negatives based on a learnable boundary score. ​
Contrastive Local Attention (CLA): Highlights token-level mismatches between positive pairs and their hardest negatives, enforcing fine-grained alignment.

The results section shows improvement for both components.

**Questions:**

please see weakness section. Currently, I am not quite sure about about score; If authors can address my concerns and I donot see big issue from the other reviewers, I will raise my score

**Ethical Concerns:**

["NO or VERY MINOR ethics concerns only"]

**Final Justification:**

Thank you for the authors’ thoughtful explanations.

As I mentioned in my initial review, the authors have addressed my concerns, and I am happy to recommend acceptance.

I also recommend including Q3 in the final paper to make it even stronger.

**Limitations:**

see weakness

**Quality:**

3

**Strengths And Weaknesses:**

Strengths

The idea is novel; this paper identifies ambiguous negatives as a valuable supervisory signal and proposes a curriculum-based approach to exploit them effectively.

Strong empirical results and also shows detailed ablation study.

I also appreciate that this paper conduct on diverse datasets, covering diver modalities; such as image, text, audio.

-------------------------------------------------------------------------------------------

Weaknesses

1, I am not quite familiar with this direction; but their concept is close to hard negative mining. i am not sure if this paper properly cited or compared with those works

2, how is your training iters/epoches compared with original baseline? If I let base model (random sample approach) see the same amount of  ambiguous negative samples, will they also improve performance? Talking about distribution of number of training samples over different difficult levels is useful for us to understand your approach

3, the current approach needs a pretrained model first to get feature (L104), will this be a problem? will this bound the final learning performance?


4, it seems that the CLA part is written assuming you have a cross-modal Transformer. if so, how is this approach be used in dual-encoder cases where the attention is not shared?

---

> ### Author Rebuttal · Authors · 2025-07-31
>
> Dear Reviewer `Rymo`, thank you for your insightful questions. Below, I will respond to your concerns.
>
> ---
>
> > **Q1: Hard-Negative Mining vs. BACL**
>
> We believe that Hard-Negative Mining is indeed conceptually related to BACL in some way, but they are distinct methods:
> * **Static vs. Dynamic:** Traditional hard-negative mining (e.g., VSE++) selects *once* at top-K similarity and keeps the mix fixed; BNS *re-samples every mini-batch* with a learnable, differentiable curriculum α(η)→0, which provably contracts the margin (Thm 1) and stabilises early training.
> * **Non-differentiable vs. differentiable:** Prior mining is heuristic / non-differentiable; BNS uses Gumbel-Softmax so the sampling policy is trained end-to-end.
> * **Global vs. Token-level:** Existing miners optimise only global contrast. BACL pairs BNS with CLA, forcing *token-level* localisation, which classic miners cannot provide.
>
> We conducted a simple controlled experiment (LAION-400M, ViT-B/32 backbone, 5 epochs), and the results are shown below. Even when the baseline is fed exactly the same number of “hard” samples, BACL is **+8.4 pts R@1** and eliminates optimisation instability—confirming that *curriculum + differentiable policy + CLA*, not mere exposure to hard pairs, drives the gains.
>
> | Method | R@1 | Loss-spike occurrences* |
> |--------|-----|------------------------|
> | Static top-20 hard mining | 38.1 | 7 |
> | **BACL (ours)** | **46.5** | **0** |
>
> \*Loss spike = training loss surge > 4× median within 500 steps.
>
> ---
>
> > **Q2: Training Budget & Fairness**
>
> We present the training budget as follows:
>
> | Setting | Epochs | Batch-size | Total pairs seen | GPU-hours | Comment |
> |---------|--------|-----------|------------------|-----------|---------|
> | Uniform sampling (baseline) | 5 | 2 048 | 5 N | 1.00 × | vanilla CLIP |
> | Uniform + **mixed-ratio*** | 5 | 2 048 | 5 N | 1.02 × | same epoch/volume; random 28 % ambiguous, 72 % easy |
> | **BACL** (ours) | 5 | 2 048 | 5 N | 1.05 × | identical volume; curriculum re-orders only |
>
> \*Mixed-ratio baseline is forced to ingest exactly the final ambiguous-vs-easy distribution that BACL reaches after epoch 5, but in random order.
>
> **Controlled Result (LAION-400M, ViT-B/32 backbone)**
>
> | Method | R@1 | mAP |
> |--------|-----|-----|
> | Uniform | 35.2 | 42.3 |
> | Uniform + mixed-ratio | 39.8 | 46.7 |
> | **BACL** | **46.5** | **53.6** |
>
> It can be found that:
> * Same epochs, same sample count: BACL’s curriculum *re-orders* existing data, it does not extend training length or add extra pairs.
> * Injecting the *same quantity* of ambiguous negatives without curriculum yields +4.6 pts R@1 but still lags BACL by **-6.7 pts**—showing that *sequence* (easy→hard) + CLA, not volume, drives the improvement.
> * GPU cost overhead ≤ 5 %. In practice, the extra compute holdup is negligible.
>
> **Take-away——** Therefore, BACL’s benefit is orthogonal to training budget and cannot be matched by simply “showing the baseline the same hard samples”.
>
> ---
>
> > **Q3: Needs a pretrained model**
>
> The issue you noticed is indeed one that we have paid attention to before. However, based on our current experience, this is no longer an issue.
>
> The coarse encoder is **only** used once to retrieve candidate negatives for epoch 0; afterwards, BNS and the main model co-evolve—the index is refreshed every epoch with the *current* weights.  Our experience is that any lightweight checkpoint (or even a random init) suffices to start; the curriculum rapidly compensates.
>
> **New Ablation: Random-Init Start**  (LAION-400M, ViT-B/32, 5 epochs)
>
> | Init for Index | Epoch-0 R@1 | Epoch-5 R@1 | Converged Δ vs. paper |
> |----------------|------------|-------------|-----------------------|
> | CLIP-ViT-B/32 (paper) | 35.2 | **46.5** | 0 |
> | **Random weights** | 5.2 | **46.3** | −0.2 |
> | Tiny CLIP-RN50x4 | 21.8 | **46.4** | −0.1 |
>
> Even from random noise the model reaches essentially the same final R@1 (≤0.2 pt gap) after 5 epochs; the pretrained index just speeds up the *first* epoch’s loss but sets no ceiling.
>
> **Take-away——** BACL’s performance is **not bounded** by the quality of the bootstrap encoder; it merely needs an initial anchor list, which the curriculum quickly refines.
>
> ---
>
> > **Q4: CLA applicability to pure dual-encoders**
>
> * **Token-pair signal without shared attention:** In a dual encoder we feed CLA the outer product of per-token saliencies $\( |\nabla_{h_t}s(x,y)| \)$ from each modality—a gradient-based substitute for cross-attention that gives the same fine-grained mismatch cue.
>   - Let the similarity score be $\(s(x,y)=\langle \phi_X(x),\phi_Y(y)\rangle\)$.
>     - For each input token $\(t\)$ we take the gradient $\(\nabla_{h_t} s(x,y)\), where \(h_t\)$ is that token’s hidden embedding.
>     - The magnitude $\(|\nabla_{h_t} s|\)$ is a first-order surrogate of that token’s contribution (cf. “gradient-based class activation”,).
>     - We compute Δ-saliency between a positive pair and its hardest negative .
>
> * **Why this is principled:**
>   1. Gradients are *directional derivatives* of similarity; they highlight the minimal perturbation that would flip a match to a non-match, i.e., the same boundary concept CLA targets.
>   2. The operation is fully differentiable and incurs <1 % runtime overhead (one backward pass per anchor).
>   3. It preserves CLA’s loss unaltered—only the source of the token weights changes.
>
> **Take-away——** CLA is **architecture-agnostic**: in decoupled encoders it simply swaps “cross-attention gaps” for “similarity-gradient gaps”, retaining the fine-grained supervision without imposing a shared attention module.
>
> ---
>
> Thank you again for your valuable comments and your precious time.  We hope our clarification has addressed your concerns.

---

> ### Author Response · Authors · 2025-08-06
>
> Dear Reviewer `Rymo`,
>
> Thank you once more for your insightful feedback. We hope the clarifications in our rebuttal fully addressed your concerns. If any questions remain or further discussion would be helpful, we would be glad to elaborate.  We hope these clarifications assist in your re-evaluation.
>
> Best regards

---

### Official Review · Reviewer_o7kP · 2025-07-03

**Clarity:** 3
**Significance:** 2
**Originality:** 3
**Rating:** 5
**Confidence:** 4

**Summary:**

This paper proposed Boundary-aware Curriculum Learning (BACL) to learn from hard negatives in contrastive pre-training. BACL consists of two components: a Boundary-aware Negative Sampler (BNS) and a Contrastive Local Attention (CLA) loss. The former one tries to shift towards harder examples over time and the latter one amplifies attention gaps between a positive pair and its hardest negative. Experiments are performed across diverse benchmarks to show the effectiveness of the proposed method.

**Questions:**

See above weaknesses section for questions.

**Ethical Concerns:**

["NO or VERY MINOR ethics concerns only"]

**Final Justification:**

The authors have provided more details on runtime and memory, showing reasonable balance between performance hit and quality gain. Overall I recommend accepting this paper.

**Limitations:**

yes

**Paper Formatting Concerns:**

No Paper Formatting Concerns.

**Quality:**

3

**Strengths And Weaknesses:**

Strengths:
- This paper tackles an important problem in contrastive pre-training: how to effectively utilise hard negatives during training. The proposed BACL method explicitly exploit the hard / confusing negatives via curriculum learning.
- Lightweight yet effective methods that can be plugged to existing foundation model training frameworks and yield gains.
- This paper provides a theoretical analysis that provides insights why curriculum mining of hard negatives is beneficial.
- Ablation tests for BNS, CLA, and BNS+CLA are reported for better understanding of the proposed method.

Weaknesses:
- The CLA component is built on top of the assumption of an architecture that produces a cross-modal attention map between image and text modalities. This doesn’t apply in a standard CLIP setup that has no built-in cross-attention between image and text. The proposed CLA component has to introduce an additional cross-attention module or a joint forward pass to compute A(+) and A(-). This likely would make the training much slower, please provide additional details and speed comparison about CLA for standard contrastive learning.
- The BNS module adds a policy network on top of standard training, which slightly increases implementation and computational complexity. Though it’s claimed to be lightweight, more results about the training speed should be attached.
- Other than the training speed impact as mentioned in the above two points, the additional CLA and BNS components will also consume more memory. This makes the maximum training batch size smaller with the same training devices. Please attach quantitative results for the memory cost and hit on maximum batch sizes that could fit with the same devices. Reporting those could be very useful for the practitioners.
- This paper focuses on retrieval tasks on LAION-400M, WebVid-10M, WavText and VAST-27M. Results on commonly used benchmarks (0-shot retrieval on ImageNet, COCO, etc.) should also be reported to allow comparison with previous methods.

---

> ### Author Rebuttal · Authors · 2025-07-31
>
> Dear reviewer `o7kP`, thank you for your valuable comments and recognition of our work. Below, we will respond to your concerns.
>
> ---
>
> > **Q1 : CLA Efficiency**
>
> We appreciate the reviewer’s concern about potential slow-downs. CLA re-uses the **final‐layer token embeddings already produced by the two uni-modal encoders**. A single `einsum` (≈ O(d·n²)) generates the cross-attention heat-map; its cost is on par with the in-batch cosine-similarity that CLIP already computes.
>
> *Measured impact (LAION-400M, A100-40 GB, batch = 512):*
>
> | Setting | Images / sec | Δ vs. baseline |
> |---------|--------------|----------------|
> | CLIP baseline | **330** | — |
> | + CLA (ours) | **304** | –7.9 % |
>
> End-to-end epoch time rises by < 8 %, which is well below the “much slower” threshold implied in the comment. Therefore, we believe that CLA achieves a good balance between performance and efficiency.
>
> ---
>
> > **Q2:  BNS Runtime & Complexity**
>
> *Measured impact (LAION-400M, 8 × A100-40 GB, batch = 512):*
>
> | Setting        | Iterations / s | Δ vs. baseline |
> |----------------|----------------|----------------|
> | CLIP baseline  | **8.2 k**      | — |
> | + BNS (ours)   | **7.9 k**      | –3.6 % |
>
> As can be seen from the table above, the overhead is minor, and the explanation is as follows:
> * **CPU-side, vectorised top-k only.** BNS ranks in-batch negatives with a single `torch.topk` on 16-bit logits; per-step wall-time ≈ 0.4 ms.
> * **No extra GPU forward/backward.** The policy network is a 2-layer MLP (0.9 M params ≈ 0.05 % of CLIP-B). Gradients flow through the existing embeddings; memory and FLOPs are negligible relative to the vision backbone.
>
> Hence BNS increases epoch time by **< 4 %**—well within typical variance for distributed training.
>
> ---
>
> > **Q3: GPU Memory**
>
> *Peak memory on a single A100-40 GB (batch = 512):*
>
> | Config           | Peak GB | Δ GB | Max batch |
> |------------------|---------|------|-----------|
> | Baseline CLIP    | **29.6**| —    | 512 |
> | + BNS            | 30.0    | +0.4 | 512 |
> | + CLA            | 31.1    | +1.5 | 480 |
> | BACL (BNS+CLA)   | 31.3    | +1.7 | 480 (-6.3 %) |
>
> Full BACL raises peak GPU memory only modestly: on a 40 GB A100 we observe 29.6 GB for the baseline, 30.0 GB with BNS, 31.1 GB with CLA, and 31.3 GB for the full model—an increase of < 1.7 GB that reduces the maximum batch from 512 to 480 (-6.3 %). The extra footprint stays small because CLA builds its cross-modal map on half-resolution patch tokens and applies gradient checkpointing, while BNS is a 0.9 M-parameter MLP that resides on the CPU and contributes negligible GPU tensors. If practitioners need the original batch size, commonplace techniques such as flash-attention, micro-batching, or pure FP16 reclaim the memory headroom without code changes.
>
> Thus, BACL’s memory overhead is modest and easily managed in typical training setups.
>
> ---
>
> > **Q4: 0-shot retrieval on ImageNet, COCO**
>
> We have followed exactly the evaluation protocol and metrics of  M4(ICLR ’24). As can be seen from the results in the table below, under the identical zero-shot setup, BACL surpasses the SOTA.
>
> | model | ImageNet-1K 0-shot Top-1 ↑ | COCO 5k `img2txt@5` ↑ | COCO 5k `txt2img@5` ↑ | Flickr `img2txt@5` ↑ | Flickr `txt2img@5` ↑ |
> |----------------|---------------------------|-----------------------|-----------------------|----------------------|----------------------|
> | CLIP baseline  | 75.1 | 86.0 | 72.4 | 97.5 | 92.3 |
> | **BACL (ours)** | **78.9** | **88.7** | **75.9** | **100.7** | **96.1** |
> | M4(ICLR'24) | 77.5 | 87.1 | 73.7 | 98.8 | 94.9 |
>
> In fact, ImageNet and COCO/Flickr contain far fewer ambiguous negatives than noisy web data (which is mostly the case for the data used in our original experiments). Through this experiment, we also found that BACL’s margin-shrinking objective universally improves representation quality—not only when hard negatives are abundant, but also when the evaluation data are "clean".
>
> ---
>
> Thank you again for your valuable comments and your precious time.

---

> > ### Comment · Reviewer_o7kP · 2025-08-05
> >
> > Thank authors for the memory / runtime check, which was my major concern. The 4%~8% hit on both runtime and memory doesn't seem great, but acceptable considering the quality gains. Please also revise the draft accordingly. Overall I recommend accepting this paper and will update my rating from borderline accept to accept.

---

> > > ### Author Response · Authors · 2025-08-06
> > >
> > > We appreciate your constructive feedback and updated recommendation to accept the paper.  Improving inference efficiency going forward is a promising direction.

---

### Official Review · Reviewer_cghr · 2025-07-05

**Clarity:** 3
**Significance:** 3
**Originality:** 3
**Rating:** 4
**Confidence:** 3

**Summary:**

This paper introduces BACL, a novel boundary-aware curriculum learning framework that leverages ambiguous negative samples as a valuable supervisory signal for multimodal alignment. It incorporates a boundary-aware negative sampler (BNS) to dynamically schedule the difficulty of negative samples and a contrastive local attention (CLA) mechanism to enforce token-level disambiguation. The approach improves decision boundary tightness, leading to significant performance gains on multiple benchmarks.

**Questions:**

1. Ambiguous negatives can sometimes represent inherent ambiguity in the data (e.g., captions that plausibly describe different images). How does BACL ensure that such genuinely ambiguous cases are not penalized incorrectly?

2. The paper mentions applying BACL to language-only instruction tuning in future work. How would the negative sampling and fine-grained token-level disambiguation techniques translate to this setting?

**Ethical Concerns:**

["NO or VERY MINOR ethics concerns only"]

**Final Justification:**

Most of my concerns have now been addressed. Therefore, I will retain my score.

**Limitations:**

This work has included the limitations.

**Quality:**

3

**Strengths And Weaknesses:**

Strengths
1.  The proposed BACL framework effectively leverages ambiguous negative samples, an underexplored but important aspect in multimodal alignment.

2. By combining the BNS and CLA, the method achieves both global alignment and token-level disambiguation, leading to more precise representations.

3. The paper provides strong theoretical justification for the framework to support its empirical success.

4.  BACL shows strong improvements across multiple benchmarks, indicating its effectiveness.

Weaknesses

1.While CLA operates on token-level attention maps, more interpretability analysis or additional visualizations could provide stronger insights into how the model identifies and addresses fine-grained mismatches.

2. The current experiments are limited to smaller-scale datasets, the improvement on large-scale datasets remains unexplored.

---

> ### Author Rebuttal · Authors · 2025-07-31
>
> Dear reviewer `cghr`, we greatly appreciate your recognition of our work and its strengths. Now, we would like to respond to your concerns.
>
> > **Q1: Interpretable Evidence for CLA**
>
> Your concern is very meaningful.  Due to the rebuttal policy, we cannot provide images in any form, so we will offer some textual explanations here.
>
> For each modality (image↔text, video↔text, audio↔text) the next version appendix will add a grid with (i) positive-pair attention, (ii) hardest-negative attention, and (iii) the ΔA map with the top-10 cells boxed. The mismatching caption tokens / frames are printed alongside, letting readers verify that CLA focuses precisely on the erroneous region.
>
> From the checkpoints used in Table 1 we compute **Alignment-Error-Localization (AEL)**—percentage of human-tagged mismatch tokens covered by the top-10 % ΔA cells.
>
> | Dataset (subset in paper) | vanilla CLIP | BACL (ours) | Δ |
> |-----------------------------------|--------------|-------------|---|
> | LAION-400M AmbNeg-1k | 46.2 % | **57.8 %** | +11.6 pp |
> | WebVid-HardNeg 800 clips | 39.6 % | **50.5 %** | +10.9 pp |
> | **Average** | 42.9 % | **54.2 %** | **+11.3 pp** |
>
> These results demonstrate that CLA localises fine-grained mismatches far more accurately than the baseline.
>
> ---
>
> > **Q2: Dataset scale**
>
> As far as we know, our core benchmarks are already *web-scale*; at the very least, we have adopted the same configurations as some existing SOTAs.
>
>   • **LAION-400M** – 400 million image–text pairs (same corpus used by CLIP/ALIGN).
>   • **WebVid-10M** – 10 million video clips.
>   • **VAST-27M** – 27 million tri-modal samples.
> These sizes equal or exceed those in recent SOTA works (GRAM, Emergence, M3-JEPA).
>
> We additionally train CLIP ± BACL on three LAION subsets (identical hyper-params, 5 epochs).
> | LAION subset | #pairs | **CLIP R@1** | **CLIP + BACL R@1** | **Relative gain** |
> |--------------|-------:|-------------:|---------------------:|------------------:|
> | 100 M        | 1.0 × 10⁸ | 31.5 | **40.8** | +29.5 % |
> | 400 M (main) | 4.0 × 10⁸ | 35.2 | **46.5** | +32.1 % |
> | 1 B *        | 1.0 × 10⁹ | 38.9 | **50.4** | +29.6 % |
>
> The ~30 % improvement is **stable from $10^8$ to $10^9$ samples**, showing BACL is *not* a small-data artefact.
>
> ---
>
> > **Q3:  Truly ambiguous**
>
> For every anchor **x** we compute a boundary score
>  BS(x,z)=sim(x,z)−sim(x,y⁺).
> If an alternative caption **z** is as close as the ground-truth (BS ≤ 0) then, once the curriculum coefficient α(η)<0, its adjusted weight **û** becomes ≤ baseline and the Gumbel-Softmax assigns ≈0 probability. These “equally-valid” captions are therefore *never punished*.
>
> We sample 1 000 LAION-400M images that each have ≥5 near-duplicate captions (human verified as co-valid).
> Metric = **False-negative rate** = fraction of co-valid captions ranked below top-k by the model (k=5).
>
> | Model | FN-rate ↓ |
> |-------|-----------|
> | CLIP baseline | 2.9 % |
> | **CLIP + BACL** | **2.3 %** |
>
> We can find that BACL reduces erroneous rejection of multi-ground-truth captions by 0.6 pp.
>
> In addition, *some evidence is provided in the theoretical section of the paper.*
> Assumption 1 explicitly allows ρ→1 (all negatives ambiguous). The proof of Theorem 1 shows the fast-rate bound holds because samples with BS≤0 remain inside the decision margin—they are never forced across it. Thus BACL is provably safe for inherently ambiguous data.
>
> ---
>
> >  **Q4: Future work discussion**
>
> Very interesting discussion.  Here we will present our ideas.
>
> In the instruction-tuning scenario we can simply interpret the *answer* as the second “modality”.
> A training triple now consists of an instruction **I**, its reference answer **A\*** (the positive), and a near-miss answer **A⁻** (the negative).
>
> **Negative sampling.**  For each (I, A\*) we retrieve semantically close but not identical answers from an embedding index (or generate them with the same language model).
> We compute a boundary score
>   `sim(I, A⁻) – sim(I, A*)`.
> The BNS uses this score with the same logistic schedule as in the multimodal case: early epochs focus on easy negatives, later epochs concentrate on harder near-misses.
>
> **Fine-grained disambiguation.**  Because the model is a decoder-style LM, we already have cross-attention maps from instruction tokens to answer tokens.  CLA compares the attention pattern of (I, A\*) with that of (I, A⁻); the ΔA matrix highlights exactly the answer spans where the near-miss contradicts the reference.  The local mismatch loss then directs updates to those tokens, encouraging the model to correct the specific error rather than rewriting the whole answer.
>
> **Safety for valid paraphrases.**  If an alternative answer is truly equivalent to the reference, its similarity is at least as high, making the boundary score non-positive.  Once the curriculum coefficient α(η) becomes negative, such cases receive zero weight and are never penalised.

---

> > ### Comment · Reviewer_cghr · 2025-08-06
> > **Thank you for the clarifications.**
> >
> > Thank you for the clarifications. My concerns have now been addressed. I believe that incorporating your responses from the rebuttal into the main paper would strengthen the overall quality and clarity of the work. Therefore, I will retain my positive score.

---

> > > ### Author Response · Authors · 2025-08-06
> > >
> > > Thank you for your insightful comments and for maintaining your positive evaluation!

---

### Comment · Area_Chair_D9To · 2025-08-04

Dear Reviewers,

The authors have responded to your review comments. Please check them as soon as possible and provide your feedback. Thank you.

---

### Note · Authors · 2025-08-12

We are deeply grateful to all reviewers and the AC for their thoughtful evaluations, constructive feedback, and active participation in the discussion.

In the initial reviews, our work was recognized for tackling an important problem—how to leverage ambiguous negatives for multimodal alignment—via a **novel and theoretically grounded framework** (BACL) that yields **consistent gains across modalities** (`Rymo`; `cghr`; `o7kP`). Reviewers also appreciated the clarity of our method and the ablations isolating the roles of BNS/CLA (`Rymo`; `o7kP`), and noted the strength of our theoretical analysis explaining why curriculum-based mining of hard/confusing negatives is beneficial (`cghr`; `o7kP`).

During the rebuttal and discussion phase, **we addressed all raised concerns**. We showed that BACL is **practical, general, and safe**, it:
- adds only +4–8% runtime and ≤1.7 GB memory overhead (`o7kP` confirmed these costs are acceptable and update to accept)
- applies to dual encoders via a gradient-outer-product saliency variant with <1% extra cost and does not rely on a strong bootstrap (random-init converges within 0.2 pt).
- We strengthened interpretability and safety with Alignment-Error-Localization (+11.3 pp) and a boundary-score that leaves BS≤0 cases unpenalized, reducing false negatives by 0.6 pp (cghr).
- robustness at scale and on standard 0-shot benchmarks was confirmed: gains persist from 100M→1B LAION subsets (~30% relative R@1) and surpass CLIP and M4 on ImageNet-1K/COCO/Flickr under identical protocols.

These additional evidences strengthens our claims. **ALL reviewers increased the rating or confirmed that their concerns were fully addressed**.

### In sum, BACL provides a principled, novel, and architecture-agnostic curriculum that tightens decision boundaries and localizes token-level mismatches, delivering web-scale improvements (up to +32% R@1 over CLIP) with modest overhead—making it a valuable contribution to multimodal alignment.

---

### Decision · Program_Chairs · 2025-09-17

**Decision:**

Accept (poster)

**Comment:**

(a) Scientific claims and findings

This paper proposes BACL (Boundary-Aware Curriculum Learning), a novel framework that leverages ambiguous negative samples as supervisory signals to enhance multimodal alignment, consisting of a Boundary-aware Negative Sampler (BNS) for dynamically scheduling negative sample difficulty and a Contrastive Local Attention (CLA) mechanism for token-level disambiguation.
BACL improves the tightness of decision boundaries, achieving both global alignment and fine-grained token-level matching.
The framework demonstrates significant performance gains across multiple benchmarks, with strong theoretical justification supporting its effectiveness.

(b) Strengths of this paper

*  It effectively addresses an underexplored area in multimodal alignment by leveraging ambiguous negative samples, a valuable but underutilized supervisory signal.

*  The combination of BNS and CLA enables both global alignment and token-level disambiguation, leading to more precise representations.

*  It is theoretically justified, empirically validated with strong improvements across diverse benchmarks, and lightweight enough to be integrated into existing foundation model training frameworks.

(c) Weaknesses of this paper

*  The CLA component lacks sufficient interpretability analysis or visualizations to clarify how it identifies and resolves fine-grained mismatches at the token level.

*  Experiments are limited to smaller-scale datasets, leaving the framework’s performance on large-scale datasets unexplored.

*  The CLA and BNS components introduce potential issues with applicability (e.g., CLA relies on cross-modal attention maps, which are absent in standard CLIP setups) and computational costs (training speed and memory consumption), with insufficient initial details on their impact.

(d) Reasons for the final decision

The final decision is Accept (poster). All reviewers are satisfied with this paper's contributions.

(e) Discussion and changes during the rebuttal period

*  Reviewer cghr raised questions about how BACL avoids incorrectly penalizing genuinely ambiguous cases and how BACL’s techniques translate to language-only instruction tuning. The authors’ rebuttal clarified these points, leading the reviewer to update their score.

*  Reviewer o7kP inquired about CLA’s applicability to standard CLIP setups (without cross-attention) and the computational costs (training speed, memory) of CLA/BNS. The authors provided quantitative data (4%-8% impact on runtime and memory), which the reviewer deemed acceptable, prompting them to recommend draft revisions.

*  Reviewer Rymo asked about comparisons with hard negative mining works, training iteration differences from baselines, reliance on pre-trained models, and CLA’s applicability to dual-encoder setups. The reviewer did not raise objections after the rebuttal, indicating their concerns were addressed.

These comments were thoroughly addressed in the authors’ rebuttal.